# Observing local $CO_2$ sources using low-cost, near-surface urban monitors

Alexis A. Shusterman[1], Jinsol Kim[2], Kaitlyn J. Lieschke[1], Catherine Newman[1], Paul J. Wooldridge[1], Ronald C. Cohen[1,2]

[1]Department of Chemistry, University of California, Berkeley, Berkeley, 94720, CA, USA
[2]Department of Earth and Planetary Science, University of California, Berkeley, Berkeley, CA, 94720, USA

*Correspondence to*: Ronald C. Cohen (rccohen@berkeley.edu)

**Abstract.** Urban carbon dioxide comprises the largest fraction of anthropogenic greenhouse gas emissions but quantifying urban emissions at subnational scales is highly challenging, as numerous emission sources reside in close proximity within each topographically intricate urban dome. In attempting to better understand each individual source's contribution to the overall emission budget, there exists a large gap between activity-based emission inventories and observational constraints on integrated, regional emission estimates. Here we leverage urban $CO_2$ observations from the BErkeley Atmospheric $CO_2$ Observation Network (BEACO2N) to enhance, rather than average across or cancel out, our sensitivity to these hyperlocal emission sources. We utilize a method for isolating the local component of a $CO_2$ signal that accentuates the observed intra-urban heterogeneity and thereby increases sensitivity to mobile emissions from specific highway segments. We demonstrate a multiple linear regression analysis technique that accounts for boundary layer and wind effects and allows for the detection of changes in traffic emissions on scale with anticipated changes in vehicle fuel economy–an unprecedented level of sensitivity for low-cost sensor technologies. The ability to represent trends of policy-relevant magnitudes with a low-cost sensor network has important implications for future applications of this approach, whether as a supplement to sparser existing reference networks or as a substitute in areas where fewer resources are available.

## 1 Introduction

Initiatives to curb greenhouse gas emissions and thereby reduce the extent of climate change-related damages are gaining momentum from city to global scales (United Nations, 2015). To support this effort, there is a clear need for monitoring strategies capable of describing emission changes and attributing those changes to the relevant policy measures (Pacala et al., 2010). Currently, an estimated 70–80% of global $CO_2$ emissions are urban in origin and this fraction is expected to grow as migration to urban areas continues and intensifies with the industrialization of developing nations (United Nations, 2011). However, cities also present the largest atmospheric monitoring challenge in that many disparate emission sources combine with complex topography.

A considerable amount of emission estimation work has been invested in the development of activity-based emission inventories for selected metropolitan areas, such as Indianapolis (Gurney et al., 2012), Paris (Bréon et al., 2015), Los Angeles

(Newman et al., 2016), Salt Lake City (Patarasuk et al., 2016), and Toronto (Pugliese et al., 2018), as well as other inventories constructed and maintained by individual air management agencies for internal use. These inventories, when updated regularly, offer the possibility of direct source attribution without the use of computationally intense and/or heavily parameterized atmospheric transport models; they do, however, typically rely on interpolations, generalizations, or proxies to generate the necessary input activity data. The Fuel-based Inventory for Vehicle Emissions developed by McDonald et al. (2014), for example, uses a representative 7 days of highway traffic flow measurements to drive the weekly cycle of $CO_2$ emissions from mobile sources on roads of all sizes year round. While traffic patterns as well as residential and commercial energy usage are known to vary by day of week (Harley et al., 2005), the specific timing and magnitude of these variations are likely to be heterogeneous in space and time. Mobile emission estimates constructed using an average week of highway observations therefore neglect the impact of anomalous events as well as the variety of vehicle fleets, commute practices, and congestion patterns that occur at the neighborhood level. As knowledge of emission factors and fuel efficiency grows, activity data will become one of the largest sources of uncertainty in bottom-up inventory products.

Ambient atmospheric measurements offer the opportunity to observe nuanced variations in $CO_2$ emission activities directly without generalizing across space and time. In order to document baseline conditions in and upcoming changes to urban greenhouse gas emissions, surface-level monitoring campaigns in cities using varied approaches are being pursued (e.g., Bréon et al., 2015; Chen et al., 2016; McKain et al., 2012; McKain et al., 2015; Shusterman et al., 2016; Turnbull et al., 2015; and Verhulst et al., 2017). These networks, typically consisting of 2–15 instruments, attempt to constrain and supplement activity-based emission inventories with observation-based estimates. Most previous work on observation-based emission estimates has focused on domain-wide emission totals over monthly to annual timescales (e.g., Kort et al., 2013). This emphasis on integrated signals has led to site selection and data analysis techniques that minimize sensitivity to local emissions, thus discarding a large portion of the information contained in the datasets collected at individual measurement sites and the differences between them (Shusterman et al., 2016; Turner et al., 2016).

We hypothesize that, if trends in the specific, small-scale $CO_2$ sources implicated in most mitigation strategies are to be resolved from atmospheric monitoring datasets, site-to-site heterogeneity must be sought out and retained. Here we present an initial characterization of the degree of spatial heterogeneity present in an urban monitoring dataset and offer these direct observations of intracity heterogeneities as a possible strategy for providing direct constraints on $CO_2$ emissions from individual sectors. We provide an initial approach to quantifying changes in the mobile sector and separating the influence of that sector from other emissions.

## 2 Measurements

### 2.1 The BErkeley Atmospheric $CO_2$ Observation Network

The BErkeley Atmospheric $CO_2$ Observation Network (BEACO2N; see Shusterman et al., 2016) is an ongoing greenhouse gas and air quality monitoring campaign operating in the San Francisco Bay Area since late 2012. The current network is comprised

of ~50 "nodes" stationed on top of schools and museums at approximate 2 km-intervals (Fig. 1). The nodes contain a variety of commercially available, low-cost sensor technologies: a Vaisala CarboCap GMP343 for $CO_2$, a Shinyei PPD42NS for particulate matter, a suite of Alphasense B4 electrochemical devices for $O_3$, CO, NO, and $NO_2$, as well as meteorological sensors for pressure, temperature, and relative humidity. Data is collected every 2–10 s and transmitted wirelessly or via an onsite Ethernet connection to a central server, where it is made publicly available in near real time. The distributed low-cost dataset is supplemented by a "supersite" at the RFS location featuring a Picarro G2401 cavity ring-down spectroscopy analyzer for $CO_2$, CO, and $H_2O$, a TSI Optical Particle Sizer 3330 for particulate matter, a ThermoFisher Scientific 42i-TL $NO_x$ analyzer for NO and $NO_2$, a Teledyne 703E photometric calibrator for $O_3$, a Pandora spectrometer system for total column $O_3$ and $NO_2$, a Lufft CHM 15k ceilometer for cloud and aerosol layer height, as well as various instruments for meteorological measurements (i.e., a Vaisala WXT520 weather transmitter, a Campbell Scientific CS500 temperature and relative humidity probe, and a Davis Vantage Pro2 system with a Davis 6410 anemometer and Davis 6450 solar radiation sensor). This high-cost, reference-grade instrumentation serves as a high-accuracy anchor point within the network domain. Atmospheric boundary conditions are monitored by the Bay Area Air Quality Management District's Greenhouse Gas Measurement Program, which maintains its own reference instruments at four background sites to the northwest, east, southeast, and south. A description of the design, deployment, and evaluation of the BEACO$_2$N approach can be found in Shusterman et al. (2016) and Kim et al. (2018).

Here we utilize $CO_2$ observations from the 20 BEACO$_2$N sites operating most consistently during the summer and/or winter of 2017 (Table 1), defined as 1 June 2017 through 30 September 2017 and 1 November 2017 through 31 January 2018, respectively. The raw 2-second $CO_2$ concentrations are averaged to 1-minute means, which are subsequently converted to bias-corrected dry air mole fractions using site-specific meteorological observations and in-network reference measurements (see Shusterman et al., 2016). The processed 1-minute averages are assumed to have an instrumental uncertainty of less than ±4 ppm. The longer averaging timescales used hereafter reduce the error of the mean (e.g., ±1.8 ppm at 5-minute resolution, ±0.5 ppm at hourly resolution, ±0.06 ppm for a given hour of the day over an entire season, etc.), although the concomitant increase in the influence of atmospheric variability cannot be quantified. Any long-term drift in the sensors is accounted for via a combination of periodic (i.e., every 12–24 months) laboratory recalibration and a post hoc data treatment based on an independent reference site in the network domain. This procedure allows us to confidently compare measurements taken multiple years apart, thus enabling inter-annual changes in $CO_2$-related phenomena to be monitored. The exact details of the calibration and post hoc data treatment are provided in Shusterman et al. (2016)."

## 2.2 Traffic Counts

Traffic count data is collected by the California Department of Transportation as part of the Caltrans Performance Measurement System (PeMS; http://pems.dot.ca.gov/). Hourly passenger vehicle flow data (in vehicles per hour) are obtained from the road monitors nearest to the relevant BEACO$_2$N site with >50% directly observed (as opposed to modeled) data and are summed across all lanes and directions. Due to limited data coverage, in some cases it is necessary to sample road monitors upstream

or downstream of the desired roadway segment; here we assume the sampled traffic conditions to be reasonable approximations of those on the desired segment. The specific monitor IDs used in each analysis are given in Table 1.

## 3 Results & Discussion

To quantify the spatial heterogeneity present across the network, we examine the degree of correlation between every possible pairing of sites in a given season as a function of the distance between them, borrowing from a similar analysis used by McKain et al. (2012). For straightforward comparison with the McKain et al. results, we first average the total $CO_2$ mole fractions to 5-minute resolution. Then, for every pairwise combination of two sites, we perform an ordinary least squares linear regression between the two 5-minute time series and calculate the Pearson correlation coefficient. We repeat this procedure after offsetting the two time series by ±5 minutes, ±10 minutes, etc., allowing for up to a ±3-h lag and choose the optimal $r^2$ value from the possible offsets. We plot the thus optimized pairwise correlations as a function of the distance separating the two relevant sites (Figs. 2 and 3) and fit the results to a single term exponential decay on top of a constant background, defined by the mean correlation observed at inter-site distances greater than 20 km.

In the summer months, there appears to be some relationship between the proximity of the sites and the correlation of their observations at all hours, with higher correlations between neighboring sites decaying into more modest, but still significant, correlations at longer inter-site distances. The characteristic length scale of this correlation is 2.9 km (defined as the e-folding distance of the exponential fits in Fig. 2; 3.6 km during the day and 2.2 km at night), which we interpret as an indicator of the distance at which various emission sources exert influence over a site's measurements. Shorter correlation lengths indicate sensitivity to near-field emissions, while longer correlation lengths imply sensitivity to far-field phenomena.

The winter months exhibit lower pairwise correlations overall and shorter correlation lengths relative to the summertime (2.4 km; 2.6 km during the day and 2.1 km at night). Some portion of the summer–winter differences may be attributable to seasonal differences in dominant wind patterns, although this effect is difficult to disentangle from the slightly different collection of sites sampled during the two seasons; the winter sample, for example, contains fewer pairs with separation lengths less than 5 km, which affects the perceived overall trend. In either season, the correlation lengths are, as expected, considerably longer than the previously observed ~100 to 1000 m e-folding distances of reactive urban pollutants that are also lost via chemical pathways (e.g., Zhu et al., 2006; Beckerman et al., 2008; Choi et al., 2014), thus validating the original choice of 2 km as the desirable inter-site separation in the design of the BEACO$_2$N instrument.

The 24-hour findings (top panels of Figs. 2 and 3) compare well to those presented by McKain et al., who also documented a decaying but nevertheless persistent correlation with increasing site separation. However, McKain et al. saw very little correlation after restricting their analysis to daytime hours, even at very short (<5 km) inter-site distances, which implies that daytime observations reflect hyperlocal phenomena only. In contrast, we observe moderate to high correlations during the day, which illustrates that information about emissions and transport phenomena on a variety of scales is preserved. A spatial visualization of the daytime correlation coefficients at four representative winter sites is shown in Fig. 4. We see that PER is well correlated with its neighbors only, suggesting the presence of local phenomena that do not affect other parts of the

network. LCC, however, also exhibits relationships with more distant sites, indicating a sensitivity to more regional-scale (10–30 km) influences. Meanwhile, HRS and OHS each possess at least one near neighbor with whom they are poorly correlated, perhaps due to hyperlocal events specific to those sites. While the region-wide phenomena can be characterized using sparser networks of high-cost, conventional monitoring equipment, the ability to capture these local processes is unique to the high-density approach.

We posit that the true strength of a high-density, surface-level monitoring network lies in its characterization of hyperlocal phenomena unique to a given site or subset of sites. In order to directly examine signals attributable to these specific, local $CO_2$ emission processes, we separate each site's observations into a "regional" and "local" component. The regional component is, by definition, the same at all sites network-wide, calculated from the bottom 10th percentile of all BEACO$_2$N readings collected during the surrounding 1-hour window. The bottom 10th percentile is chosen (rather than the absolute minimum) to account for measurement error (±4 ppm at 1-min resolution; see Shusterman et al., 2016) as well as any near-field draw down from the local biosphere; negative values in the local signals are likely attributable to some combination of these effects. While many different sites contribute to this bottom 10th percentile over the course of the data record, some sites located in close proximity to emission sources are never represented in the bottom 10th percentile and always exhibit some enhancement (i.e., a non-zero local component) over the regional background signal. The regional component is allowed to vary throughout the data record and will therefore reflect domain-wide changes in response to day of week, synoptic weather events, etc.

The diel profiles of the regional signal measured in summer and winter 2017 are shown in Fig. 5, reflecting the typical convolution of background concentrations, emission processes, and dynamics experienced across the entire BEACO$_2$N domain. In both seasons, we see an increase in the regional signal beginning around 0400 local time (LT), followed by a decrease in concentrations at 0800 LT in the winter months and 1100 LT in the summer, and another increase in early to late afternoon depending on the season. This diurnal profile corresponds well with known patterns in traffic emissions–which are largely consistent across seasons–superimposed on diel fluctuations in boundary layer height and/or biosphere activity that vary in timing and magnitude according to the season. Namely, these results might be interpreted to conclude the nighttime boundary layer in the BEACO$_2$N domain to be shallower during the winter months, producing a larger regional increase in response to rush hour traffic. The wintertime layer also appears to expand and re-contract earlier in the day than the summertime layer, resulting in both an earlier minimum and an earlier rise in afternoon–evening concentrations. The larger amplitude of the wintertime diurnal cycle may also reflect the greater influence of daytime photosynthesis and nighttime respiration during the San Francisco Bay Area's rainy winter season. An analysis of the regional signals calculated for similar periods in 2013 revealed qualitatively similar results (Fig. S1), although it should be noted that the 2013 analysis uses observations from a significantly different subset of sites in the BEACO$_2$N network.

We isolate the local signals by subtracting the network-wide regional component from the data record at each site. Median 1-minute local $CO_2$ signals range from 0.3 to 40.2 ppm during the day (1100–1800 LT) and 1.1 to 38.5 ppm at night (2100–0400 LT) during the summer months, although the distributions are skewed, with the 10th to 90th percentile ranges stretching

from -2.4 to 69.0 ppm during the day and -2.0 to 45.0 ppm at night. During the winter months, the daytime medians range from 3.6 ppm to 34.8 ppm (-7.0 to 90.8 ppm 10th to 90th percentile range) and -0.8 ppm to 58.7 ppm (-15.0 to 90.6 ppm 10th to 90th percentile range) at night. A full picture of the overall distributions is shown in Figs. S2 and S3, confirming a much greater frequency of high $CO_2$ concentrations during the winter months. In both seasons, the distribution of the local enhancements is typically unimodal with a heavy right-hand tail, although some sites exhibit more complex bi- or multi-modal distributions.

By definition, we expect these local signals to represent a unique combination of emission sources and atmospheric dynamics specific to a given site. Here we endeavor to determine whether measurements of local $CO_2$ enhancements can be used to monitor a single urban emission source, despite the complex landscape of $CO_2$ sources and sinks present within the student domain. We choose to focus on mobile $CO_2$ emissions as these are estimated to comprise approximately 40% of the San Francisco Bay Area's annual $CO_2$ emissions (Claire et al., 2015). This is the largest source sector in the $CO_2$ emission inventory and likely to represent an even larger fraction within the urban core, where the next largest source sectors (industrial/commercial and electricity/co-generation) are less abundant. However, as noted in the discussion of the regional signals above, direct observation of the magnitude and variation of traffic emissions via ambient $CO_2$ concentrations is complicated by the coincident variation in turbulent mixing and boundary layer height as the earth's surface warms and cools at sunrise and sunset (Fig. S4).

In order to more directly examine the relationship between highway traffic flow and urban $CO_2$ concentrations, we begin by analyzing the subset of observations collected between 0400 and 0800 LT at the LAN site, located less than 40 m from Interstate 880. During this period, traffic emissions are high, but the boundary layer is relatively shallow, thus increasing the sensitivity of the surface-level monitor to the traffic signal. The resultant strong positive correlation between rush hour traffic flow and local $CO_2$ concentrations is shown in Fig. 6. An alternative analysis using traffic density–obtained by dividing the traffic flow by the average vehicle speed–yields almost identical results (Fig. S5), revealing a factor of 2 increase in local $CO_2$ mole fraction enhancements during congestion (high traffic flow/density) relative to free-flowing conditions (low traffic flow/density), similar to that observed by a previous on-road mobile monitoring study by Maness et al. (2015). Also shown in Fig. 6 are the median $CO_2$ concentrations observed in each 500 veh h$^{-1}$ traffic flow increment and the ordinary least squares linear regression through these binned medians.

In addition to this first-order sensitivity to vehicle emissions at the near-roadway LAN site, we find that relatively subtle emission changes can also be detected using nodes stationed greater distances from the highway by controlling for the confounding impacts of dispersion and the biosphere. To do so, we decompose the $CO_2$ signals into terms that represent the influence of meteorology (which is correlated with both dispersion and biosphere activity) and emissions separately via a multiple linear regression approach analogous to that described by de Foy (2018). Briefly, we use an ordinary least squares linear regression to calculate the best fit of the relationship between a site's $CO_2$ signal and temperature, specific humidity, wind, boundary layer height, time of day, day of week, and time of year. Hourly measurements of temperature, specific humidity, wind speed, and wind direction are taken from a single NOAA Integrated Surface Database weather station at the

Port of Oakland International Airport (https://www.ncdc.noaa.gov/isd/) and 3-hour boundary layer heights are provided at 0.125º by 0.125º resolution by the ECMWF's ERA-Interim model (Dee et al., 2011; http://apps.ecmwf.int/datasets/). Although the low spatio-temporal resolution of these datasets limits their ability to capture hyperlocal meteorologies, here we follow the example of de Foy, who was nonetheless able to derive meaningful results from similarly coarse weather products.

The nonlinear relationship between $CO_2$ concentrations and wind or boundary layer height is captured by dividing these meteorological datasets into quartiles and assigning each observation a value between 0 (at the maximum of the quartile) and 1 (at the minimum) using piecewise linear interpolation. The wind speed quartiles are further subdivided by wind direction and reassigned values of 0–1 accordingly before fitting a linear coefficient to each subset. The time of year is represented as a sum of sines and cosines with annual or semiannual periodicities whose values also vary between 0 and 1 and whose amplitudes

are determined by the linear regression. Zeroes and ones are used to designate each hour of each type of day of the week as well. For example, timesteps corresponding to 0800 LT on a Monday may be assigned a 1 while all other timesteps are set to zero before the linear regression is performed. As a result, the MLR factors derived for each of the preceding explanatory variables can be interpreted in units of ppm $CO_2$. Meanwhile, the temperature and specific humidity variables are treated by calculating their difference from their mean values and dividing by their respective standard deviations before each is fit to

$CO_2$ with a single linear coefficient, which will have units of ppm $K^{-1}$ and ppm $(kg_{water}\ kg_{air}^{-1})^{-1}$, respectively.

     The independent variable leading to the greatest square of the Pearson correlation coefficient is then combined with each of the remaining variables and a second regression is performed. The two-input combination leading to the largest increase in the correlation coefficient is then combined with each of the remaining variables, and so on, until the addition of a new independent variable no longer increases the $r^2$ value by at least 0.005.

For this analysis, we use hourly total $CO_2$ concentrations (the sum of the local and regional components) measured at five sites between 15 February 2017 and 15 February 2018. For each site, the optimal set of explanatory variables and their relative contributions to the correlation coefficient are given in Table 2. Summing the products of each of the MLR factors with their respective independent variables (e.g., time of day, wind speed, etc.) gives the mixing ratio predicted by the MLR model; a representative week of observed and modeled $CO_2$ concentrations is shown in Fig. 7. We find generally good agreement, with

some significant hour-by-hour model–observation differences, especially at RFS. These do not, however, appear to be systematic either in sign or in timing (e.g., the rush hour peak in $CO_2$ may be poorly modeled on one day but well predicted on another).

     Multiple linear regression coefficients are derived for each hour of the day during five types of days of the week (Mondays, Tuesdays through Thursdays, Fridays, Saturdays, and Sundays); for clarity, Fig. 8 shows the regression coefficients for

Tuesdays through Thursdays and Sundays. Other days of the week are shown in Fig. S6. These MLR "factors" signify the average $CO_2$ enhancement or depletion (in ppm) uniquely associated with a particular hour of a particular day of the week. The dependencies on time of day and day of week derived via this method are hypothesized to primarily reflect the changes in emissions, as the influence of the coincident changes in atmospheric dynamics has been at least partially controlled for. For reference, the corresponding Tuesday–Thursday and Sunday diel cycles in the total $CO_2$ observed at each site are shown in

Fig. 9. Indeed, we do observe some of the same intuitive patterns in the linear regression coefficients, such as higher coefficients on weekday mornings corresponding to higher rush hour traffic emissions on those days, but with greater opportunity to differentiate between days of the week, especially around noon when raw concentrations are generally similar. As expected, the Tuesday–Thursday enhancement in the MLR factors is larger at the sites located close to a freeway (e.g., up to 520% higher than the corresponding Sunday MLR factor at FTK) but is less pronounced at LBL (70%), which is farther away from major mobile sources. For reference, the 1 km by 1 km FIVE mobile emission inventory developed for the San Francisco Bay Area by McDonald et al. (2014) predicts a ~210% weekday enhancement on average, peaking around 0500 LT, much earlier in the day than is observed here.

When we examine the relationship between these multiple linear regression coefficients and morning traffic flow as we did at LAN (Fig. 10), we again find positive correlations. This is an interesting result, given that the traffic flow measured on a single highway likely provides only a first order approximation of the total traffic emissions influencing a single $CO_2$ monitor, especially those situated at greater distances from said highway, which may be sensitive to additional highways, as well as local roads. Although the predominance of a single highway's emissions (or at least its correlation with those from other sources) is not a necessary condition of our MLR analysis, the strong positive correlations we observe suggest that this methodology may nonetheless be useful in monitoring emissions from individual highways such as these.

The standard error of the slope of the linear regression is calculated as the standard deviation of the model–data $CO_2$ residuals divided by the square root of the sum of the squared differences between each traffic flow increment and the mean traffic flow. The 1σ uncertainty in the slopes (i.e., the 68% confidence interval, assuming a Gaussian error distribution) is thus found to be 11–30%, indicating that analysis of a single site could be used to detect as small as 11% changes in average emissions per vehicle, an improvement upon the 17% slope uncertainty calculated for the near-highway LAN site. For reference, under the Corporate Average Fuel Economy standards, the state of California aims to achieve a fleet-wide average fuel economy of 23.2 km per liter by the year 2025 (US EPA, 2012), corresponding to a 35% decrease in emissions relative to the 15.1 km per liter economy of 2012–2016 model year vehicles. Assuming a steady decrease in emissions of 3.5% per year, an 11% decrease would be achieved after approximately 3 years, showing that one BEACO2N site is therefore sufficiently sensitive to detect such a trend with 68% confidence in as little as 3 years. By leveraging multiple independent sites, even greater confidence and/or shorter timescales could be achieved.

It is likely that even greater sensitivity could be achieved with more accurate meteorological datasets. While the single weather station and relatively coarse (0.125° by 0.125°) reanalysis product we use here may be adequate to represent the meteorological conditions across some domains, the San Francisco Bay Area is at the high end of complexity in terms of terrain and microclimatology. Higher resolution boundary layer heights and neighborhood-specific wind observations may improve the results of our multiple linear regression, but these types of measurements are rarely available on the spatial scale of the BEACO2N instrument and are difficult to simulate with accuracy (Jiménez et al., 2013; Banks et al., 2016). In future work, high-density networks like BEACO2N may therefore be useful not just in source attribution but also in providing a much needed observational constraint on our understanding of near-surface transport.

Future work will also make use of the ancillary datasets provided by the BEACO$_2$N platform, such as the concurrent NO$_x$ and CO concentrations. Prior studies have demonstrated a methodology for detecting plume-like events in the BEACO$_2$N NO$_x$ and CO observations (Kim et al., 2018), and the ratio of these species to CO$_2$ provides a unique signature for each different CO$_2$ source (e.g., Ban-Weiss et al., 2008; Harley et al., 2005; Lopez et al., 2013; Nathan et al., 2018; Turnbull et al., 2015), allowing subsets of the data record to be directly attributed to specific (e.g., mobile) source types and allowing the relationship between these specific activities and CO$_2$ mixing ratios to be derived more precisely. With such a precise methodology for converting between emissions and concentrations, subtler inter-annual trends in emissions could be detected, for example changes in vehicle emissions following construction of new housing.

## 4 Conclusions

We have described the heterogeneity measured at the individual sites of a high-density, surface-level urban CO$_2$ monitoring network. Networkwide, correlation length scales are found to be slightly longer during daytime during the summer, and generally shorter during winter months, but falling in the range of values reported previously based on other stationary observation networks and mobile monitoring campaigns. High near-field correlations are thought to be driven by shared sensitivity to local emission events, while moderate far-field correlations reflect regional episodes, suggesting that a given site's data record is likely a convolution of both phenomena. We therefore present a methodology for separating the observed CO$_2$ concentrations into local and regional components and observe distinct distributions (i.e., unimodal vs. bimodal) of local CO$_2$ enhancements within single neighborhoods. A clear relationship is seen between morning rush hour traffic counts and local CO$_2$ concentrations, allowing for the detection of changes in vehicle emissions within 2–3 years, if those changes proceed at a rate consistent with policy objectives.

Most prior studies of urban CO$_2$ emissions (e.g., McKain et al., 2012; Kort et al., 2013; Wu et al., 2018) have favored sparser networks of high-quality instruments, finding this approach to be better suited for resolving trends in total region-wide emissions. It is hypothesized that the ideal monitoring strategy depends on the particular goals and location of a given application, with certain locales and emission sources necessitating high-cost, low-density instrumentation, complemented by other domains where low-cost, high-density platforms are more effective. The potential trade-offs between measurement quality and instrument quantity specific to the San Francisco Bay Area have been investigated previously using an ensemble of observing system simulations by Turner et al. (2016), who found BEACO$_2$N-like observing systems to outperform smaller, higher quality networks in estimating regional as well as more localized emission phenomena there. While Turner et al. saw significant benefits to achieving an hourly instrument precision of 1 ppm, further increases in measurement quality offered little advantage in constraining emissions, especially those from line and point sources.

This work thus provides an important data-based validation of the conclusions of Turner et al.'s theoretical analysis. Not only do we demonstrate the ability of low-cost sensors to sufficiently constrain policy-relevant trends in line source (i.e., highway traffic) emissions, but we do so without the use of computationally intense and heavily parameterized atmospheric transport models. Furthermore, we show that a multiple linear regression analysis allows the signature of highway traffic to be

extracted from sites located throughout the network, enabling trends in mobile emissions to be quantified without specially situated, roadside monitors. Although this approach requires real time traffic count information that is not yet available in all locations, our finding is nonetheless an important result, as deriving and implementing a particular, a priori network layout is a non-trivial task. Domain-specific transport patterns prevent the development of general principles of optimal sensor placement, and, even if ideal locations can be identified, cooperation from facilities in the area cannot be guaranteed. By establishing for the first time that an ad hoc, opportunistic sensor siting approach can nonetheless provide sensitivity to emission sources of interest, we thus improve the prospects for widespread adoption of distributed monitoring systems in the future.

Progress toward evaluating the capabilities and proper use of low-cost sensors has particular relevance for nations with rapidly developing economies, where $CO_2$ emissions are increasing much faster than the resources needed to monitor them by conventional means. Domestically, citizen science and environmental justice groups are also adopting these technologies (Snyder et al., 2013) as an economically accessible means of advocating for greater public health and ecological wellbeing. While the specific correlation lengths and emission estimates we derive here are unique to the San Francisco Bay Area domain, the sensor performance capabilities and data analysis techniques we outline provide guidance more generally to any future studies attempting to interpret similar datasets around the world. High-resolution surface networks enabled by low-cost technologies offer a unique opportunity to provide ground truth constraints on difficult-to-model near-surface dynamics as well as on the individual $CO_2$ sources and sinks that comprise the strategic backbone of greenhouse gas mitigation regulation.

## 5 Data Availability

All BEACO$_2$N $CO_2$ observations used in this analysis can be downloaded at doi:10.5281/zenodo.1206983. Traffic counts are available on the California Department of Transportation website (http://pems.dot.ca.gov/), wind, temperature, and humidity observations are available on the NOAA Integrated Surface Database website (http://www.ncdc.noaa.gov/isd/), and boundary layer heights are available on the ECMWF website (http://apps.ecmwf.int/datasets/).

*Acknowledgements.* This work was funded by the National Science Foundation (1035050; 1038191), the National Aeronautics and Aerospace Administration (NAS2-03144), the Bay Area Air Quality Management District (2013.145), and the Environmental Defense Fund. Additional support was provided by a NSF Graduate Research Fellowship to AAS, a Kwanjeong Lee Chonghwan Educational Fellowship to JK, and a Hellman Fund Fellowship to KJL. We acknowledge the use of datasets maintained by the California Department of Transportation, the National Oceanic and Atmospheric Administration, as well as the European Centre for Medium-Range Weather Forecasts.

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

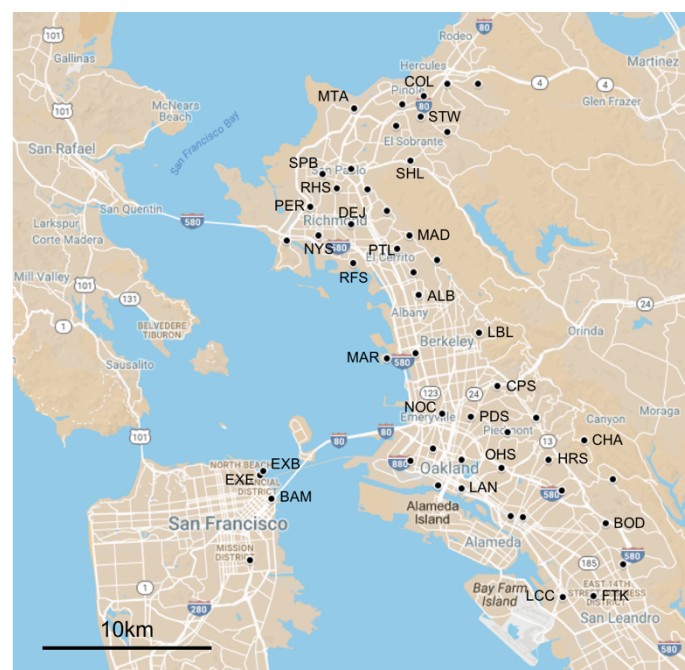

**Figure 1. Map of BEACO₂N node locations (black dots). Nodes used in this study are labeled. Map data © 2017 Google**

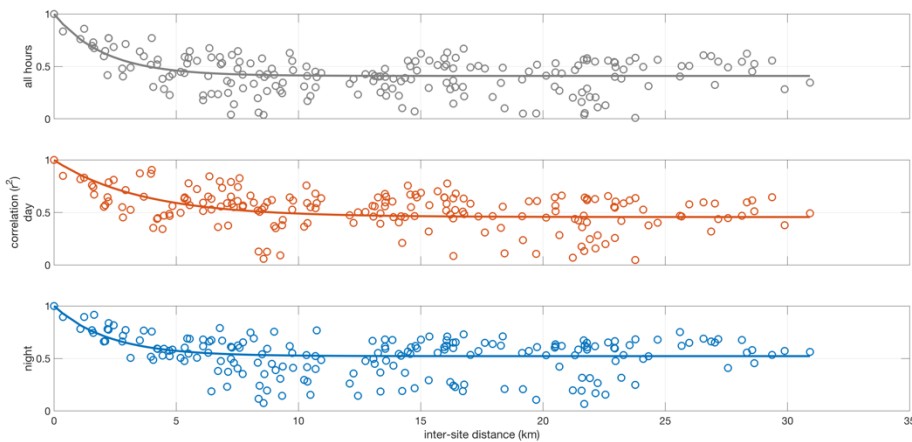

**Figure 2. Optimal correlation coefficients for every possible pairing of summer 2017 sites as a function of their separation distance during all hours (top), daytime hours (1100–1800 LT, middle), and nighttime hours (2100–0400 LT, bottom). Solid lines show exponential decay of the correlation coefficients.**

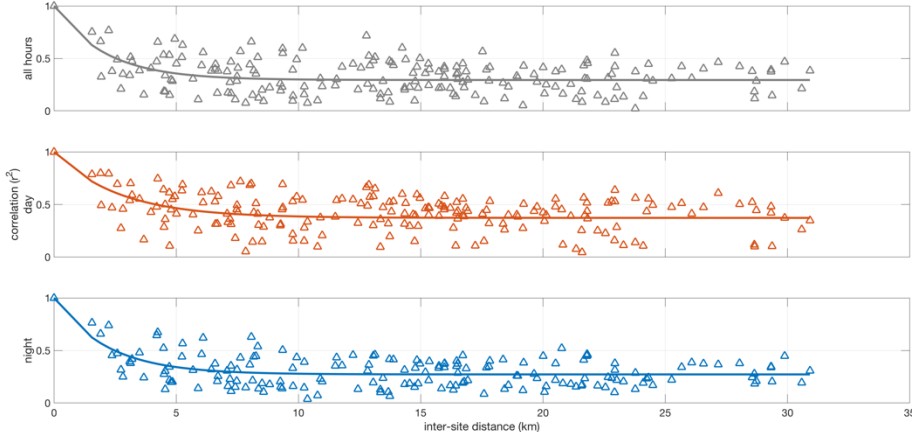

**Figure 3. Optimal correlation coefficients for every possible pairing of winter 2017 sites as a function of their separation distance during all hours (top), daytime hours (1100–1800 LT, middle), and nighttime hours (2100–0400 LT, bottom). Solid lines show exponential decay of the correlation coefficients.**

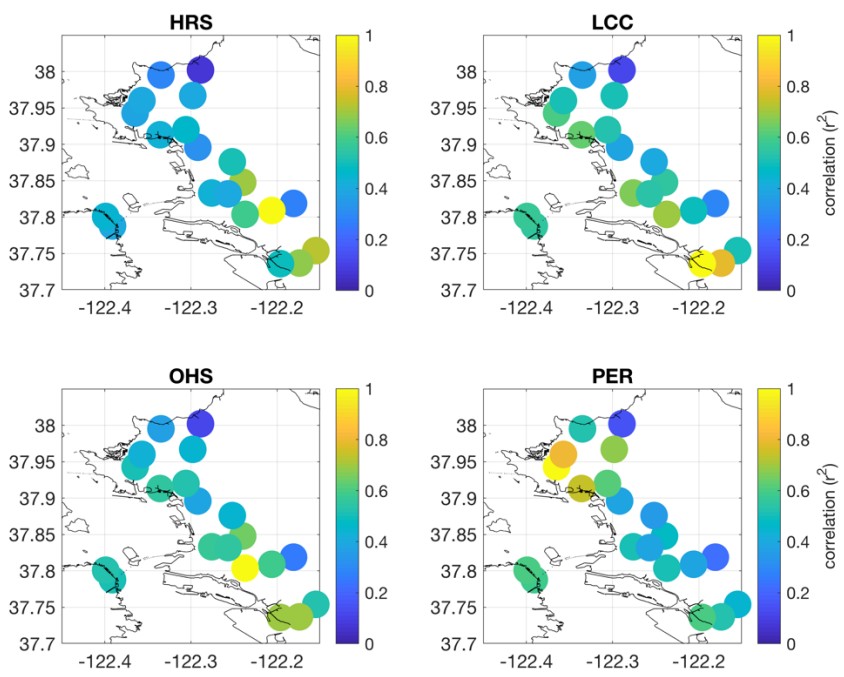

**Figure 4. Optimal correlation coefficients representing network-wide correlation with 5-minute mean total $CO_2$ concentrations at four representative sites during daytime hours (1100–1800 LT) of winter 2017. Yellow spot ($r^2 = 1$) on each subplot shows the location of the site with which the correlation is examined.**

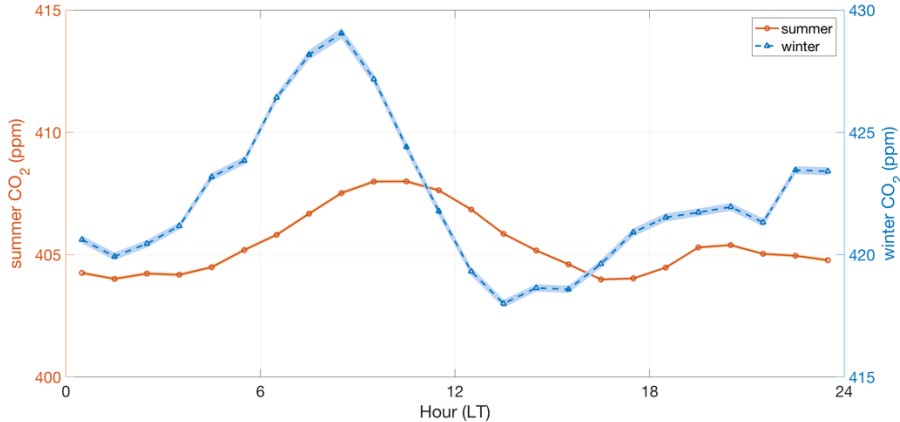

**Figure 5. Hourly median values of the network-wide, regional CO$_2$ signals calculated for summer (orange) and winter (blue) periods in 2017. Lighter colored curves indicate the standard error; note the difference in y-scale.**

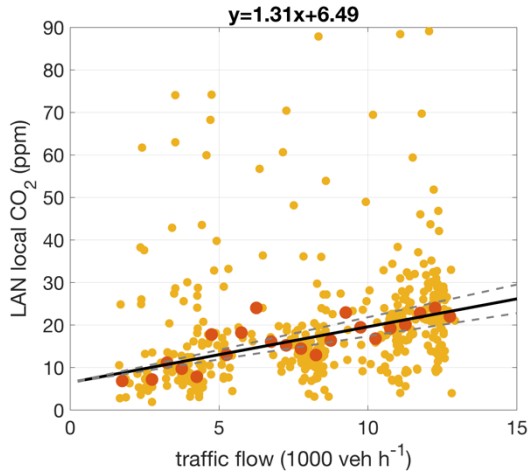

**Figure 6. Morning (0400–0800 LT) local summertime CO$_2$ concentrations at LAN shown as a function of nearby highway traffic flow. Darker points indicate the median CO$_2$ concentration observed in each 500 veh h$^{-1}$ traffic flow increment; black solid line indicates the linear regression through the binned medians (equation given above plot) and gray dashed lines show the uncertainty in the regression slope.**

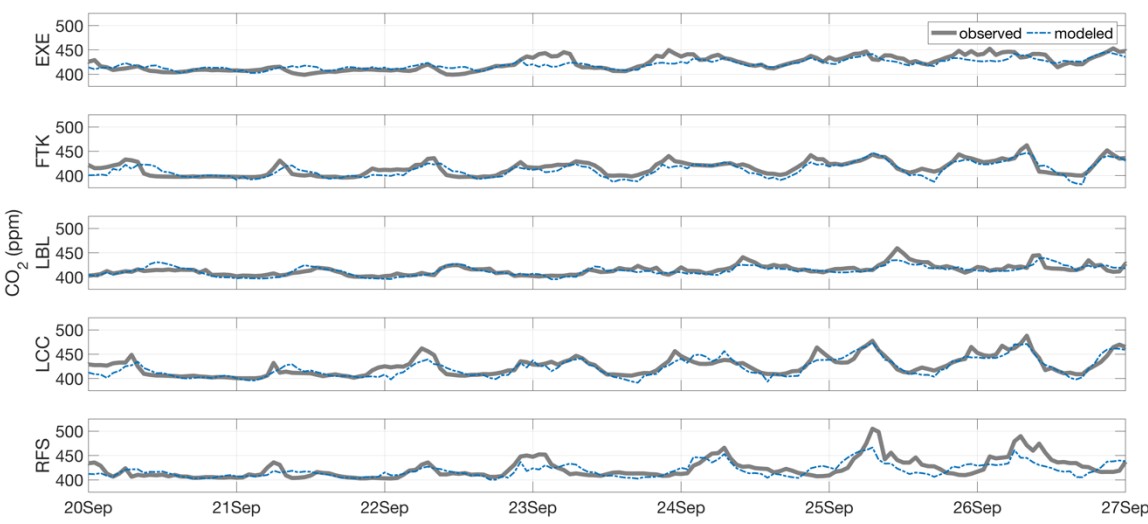

**Figure 7. Representative week of total CO$_2$ concentrations observed (thick gray curve) and modeled (dashed blue curve) at five sites using a multiple linear regression approach based on de Foy (2018).**

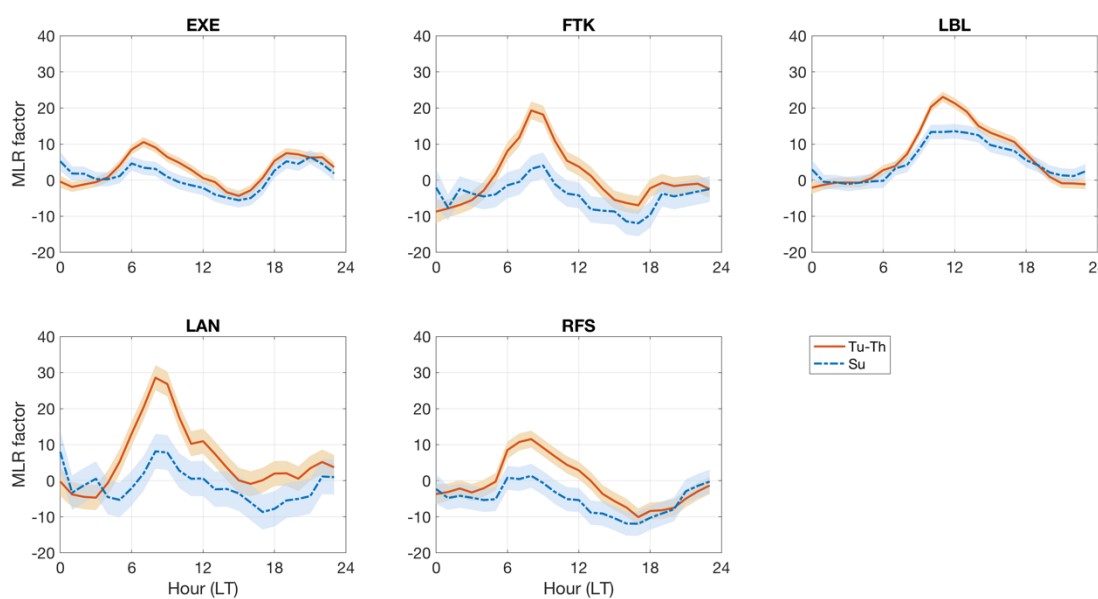

**Figure 8. Multiple linear regression coefficients for five sites derived for each hour of the day on Tuesdays through Thursdays (orange solid line) and Sundays (blue dashed line) between 15 February 2017 and 15 February 2018.**

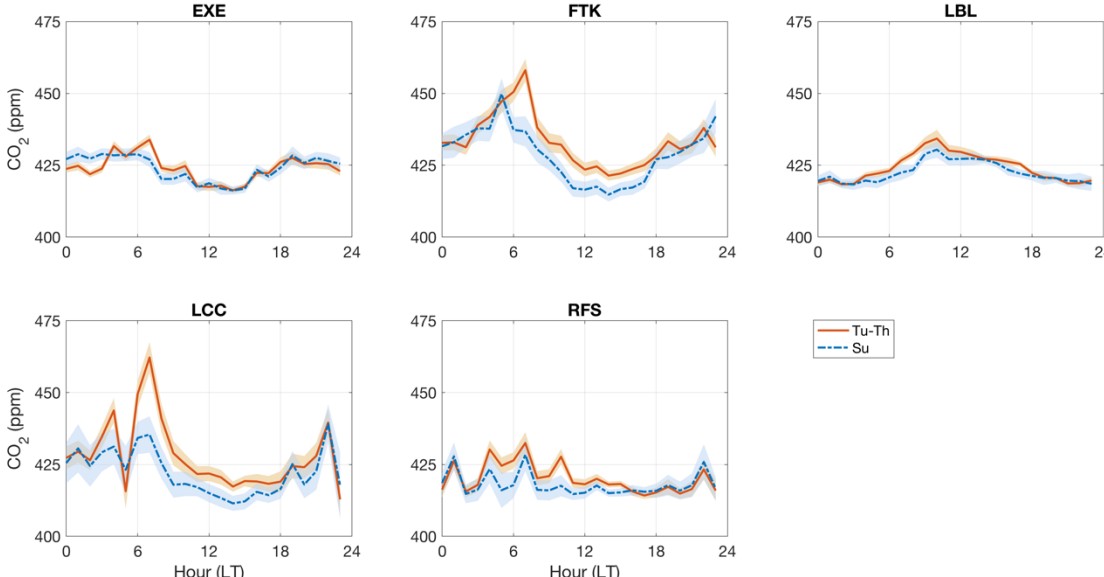

**Figure 9. Hourly median CO$_2$ concentrations observed at five sites on Tuesdays through Thursdays (orange solid line) and Sundays (blue dashed line) between 15 February 2017 and 15 February 2018; lighter curves indicate the standard error in the medians.**

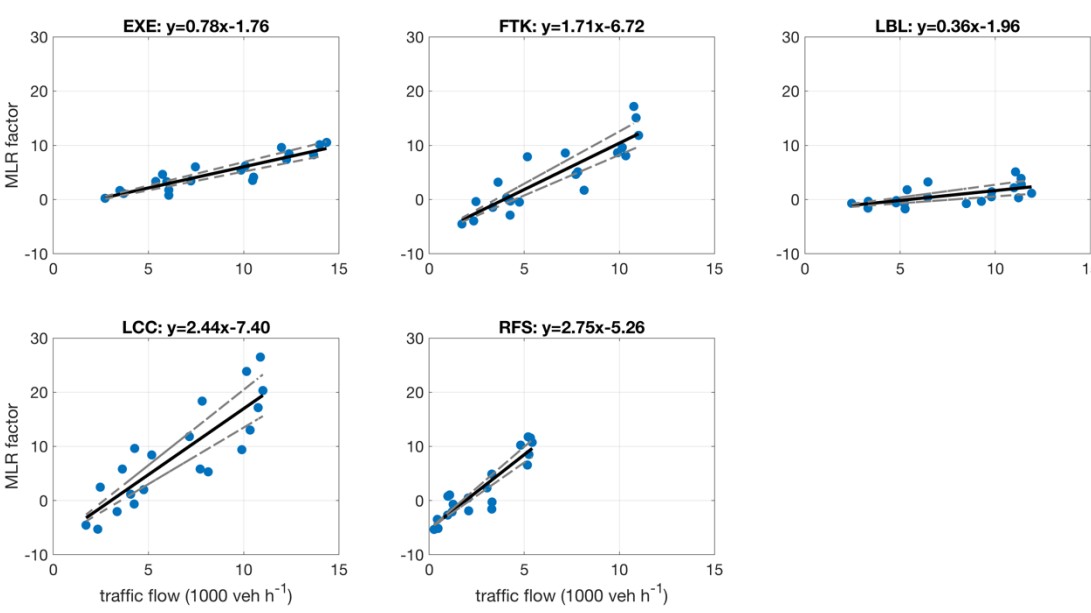

**Figure 10. Morning (0400–0800 LT) multiple linear regression coefficients shown as a function of summertime traffic flow; black solid lines indicate the linear regression through the MLR factors (equations given above each subplot) and gray dashed lines show the uncertainty in the regression slope.**

| SITE CODE | LAT (º N) | LON (º E) | TRAFFIC MONITOR IDs | DISTANCE FROM HIGHWAY (m) |
|---|---|---|---|---|
| ALB* | 37.896 | -122.292 | 401052, 402062 | 1390 |
| BAM | 37.788 | -122.391 | 402815, 404920 | 170 |
| BOD* | 37.754 | -122.156 | 401857, 401858 | 300 |
| CHA | 37.819 | -122.181 | 400302, 400308 | 1720 |
| COL | 38.002 | -122.289 | 401230, 401269 | 510 |
| CPS* | 37.848 | -122.240 | 402201, 402202 | 220 |
| DEJ† | 37.933 | -122.338 | 400361, 400445 | 950 |
| EXB† | 37.802 | -122.397 | 402815, 404920 | 1570 |
| EXE | 37.801 | -122.399 | 402815, 404920 | 1580 |
| FTK | 37.737 | -122.173 | JJAS: 400442, 400955 NDJ: 400608, 400793 | 1350 |
| HRS* | 37.809 | -122.205 | 400302, 400308 | 700 |
| LAN† | 37.794 | -122.263 | 400835, 408138 | 40 |
| LBL | 37.876 | -122.252 | 400176, 400728 | 3090 |
| LCC | 37.736 | -122.196 | JJAS: 400442, 400955 NDJ: 400608, 400793 | 220 |
| MAD† | 37.928 | -122.299 | 400819, 401558 | 1850 |
| MAR† | 37.863 | -122.314 | 400176, 400728 | 950 |
| MTA | 37.995 | -122.335 | 400538, 400976 | 2040 |
| NOC* | 37.833 | -122.276 | 401211, 401513 | 750 |
| NYS† | 37.928 | -122.359 | 400359, 400734 | 380 |
| OHS* | 37.804 | -122.236 | 400261, 401017 | 160 |
| PDS* | 37.831 | -122.257 | 400224, 401381 | 800 |
| PER | 37.943 | -122.365 | 400639, 400738 | 1790 |
| PTL | 37.920 | -122.306 | 400819, 401588 | 970 |
| RFS | 37.913 | -122.336 | 400202, 400675 | 760 |
| RHS† | 37.953 | -122.347 | 401228, 406660 | 1530 |
| SHL | 37.967 | -122.298 | 416774, 416790 | 2030 |
| SPB* | 37.960 | -122.357 | 401894, 401895 | 2280 |
| STW† | 37.990 | -122.291 | 400313, 400902 | 500 |

**Table 1. List of site geo-coordinates, relevant traffic monitor IDs, and approximate distances from a highway. Asterisks indicate sites with data available in winter 2017 only; daggers indicate sites with data available in summer 2017 only.**

| SITE | MLR VARIABLE | | | | | |
|------|--------------|--------------|------|------|------|----------|
| CODE | TIME OF YEAR | DAY OF WEEK | BLH | WIND | T | HUMIDITY |
| EXE | 0.271 | 0.031 | 0.130 | 0.028 | 0.031 | -- |
| FTK | 0.413 | 0.050 | 0.088 | 0.024 | -- | 0.010 |
| LBL | 0.230 | 0.156 | 0.052 | 0.028 | 0.008 | 0.024 |
| LCC | 0.353 | 0.062 | 0.177 | 0.021 | -- | -- |
| RFS | 0.316 | 0.052 | 0.082 | 0.029 | 0.006 | -- |

**Table 2. Explanatory variables included in the multiple linear regression analysis of each site; values indicate the correlation coefficient increase achieved by subsequent inclusion of each variable.**