# Peer review of "Observing local CO2 sources using low-cost, near-surface urban monitors"

_Atmospheric Chemistry and Physics, 2018_

## Referee Comment (RC1) · J. Turnbull (Referee) · 19 Jun 2018

This paper uses a set of observations from low-cost, near-surface sensors to examine relationships between CO2 mole fraction and traffic counts, showing strong relationships at individual sites. They also investigate correlation length scales between different sites, and develop a multiple linear regression method to establish relationships between CO2 mole fraction and the factors that influence it. The methods described appear to give exciting results showing that traffic fuel efficiency can be monitored by combining these methods with traffic count/flow information.

The data and results in this paper are interesting and entirely appropriate for publication in ACP. The major flaw in this paper is that insufficient detail of the methods is given,

and mostly only higher level data products (correlation coefficients, MLR coefficients) are given in most places. There are several instances where methods first described in other papers are here described too briefly to be understood without detailed reading of the previous publications. In other cases, figures that are key to understanding the methodology are given only in the supplementary material, and details of the "raw" information that goes into the main figures are lacking. Thus it is difficult to evaluate the robustness of the methods, and readers will have difficulty repeating the analysis or trying it out themselves. ACP doesn't have major length limitations, so the main text should be expanded so that the methodology can be followed. Specific instances are noted in my further comments.

The concepts, data and written language are all good, but I recommend major revisions to expand the explanations of the methodology and show more of the CO2 measurements and comparison to the MLR model. This will allow the reviewers and readers to better evaluate the robustness of the methodology.

Specific comments:

The effect of low-precision measurements is not detailed anywhere in this paper. How does the 0.5 ppm precision impact the results discussed? How is drift in the sensors accounted for, and how will this impact the results, particularly the concept that changes in traffic fuel efficiency could be monitored over time?

Although the authors acknowledge that traffic contributes only 40% of CO2 emissions, they then focus on only traffic emissions in the analysis. While sites very close to major roads will indeed be strongly influenced by the proximal road, sites further from roads will be influenced by multiple roads as well as the other anthropogenic sources, AND by biogenic CO2 sources and sinks. How are these other sources considered? If they are ignored in this analysis, please justify why.

Pg 1 line 25. The % of global CO2 emissions from urban areas varies depending on how it is determined . 70 to 80 % is probably a better estimate.

Section 2.2. Traffic counts. For the sites very close to a particular highway, traffic monitor data for that nearby highway makes sense. For sites that are further from any particular highway, even if traffic is the dominant proximal CO2 source, surely more than one highway (and local roads as well) will contribute to the signal observed at that site. How are multiple sources accounted for?

Pg 3 lines 31 -32, pg 4 lines 1-3. Please expand to explain the methodology used here in enough detail to be followed without requiring the reader to refer to the McKain paper. Are these correlation lengths determined using CO2 mole fraction, or the enhancements in CO2 relative to background? Where is the raw data that is used to derive these correlation lengths? Plots of the CO2 time series should be included (these could go in supplementary material).

Pg 4 lines 6 – 14. Please explain what the correlation lengths should be interpreted to mean. I take it that a shorter correlation length implies more influence of sources close to the sites. Longer length scales would imply more influence of sources further away? The preferenced studies are all about pollutant gases, not CO2 – it might be reasonable to expect higher correlations and longer length scales for a long-lived gas like CO2 with large and varying background.

Pg 4 lines 18 -19. I don't follow why the daytime correlations imply this information. Please clarify.

Pg 4 line 33. Figure 4, not figure 2?

Pg 5 lines 1 – 10. It is curious that the amplitude of the diurnal cycle is larger in winter than in summer. The timing of the diurnal pattern shown in figure 4 doesn't quite gel with the argument that this is due to lower/stronger daytime boundary layers. Have the authors considered the influence of biogenic CO2 fluxes, which may be more important in the rainy winter season in San Francisco than in the summer?

Pg 5 lines 26 – 35. I pity those commuters who are contributing to high traffic flows at

4 am!

"An alternative analysis using traffic density..." I think I understand that this is an attempt to examine how congested vs free-flowing traffic might change the results? Please clarify.

"We observe a factor of 2 difference in local $CO_2$ between congested vs free-flowing conditions" – which is higher?

How is the regression slope determined, and how is the uncertainty determined? This needs further detail, particularly because the regression slope is determined not from the full dataset, but from a fit to the median values. The idea that trends in fuel efficiency can be tracked by this method is tantalizing, but the statistics must be demonstrated to be robust.

Pg 6 lines 5-17. Not enough information is given to understand how these MLRs are constructed and therefore how they can be interpreted. Please expand on the method, and provide further details on which factors were most important. On the following page (lines 5-12), there is discussion about how improved resolution of the meteorological datasets would help, but nowhere is the current resolution and limitations of the data explained!

It isn't clear how the "modelled" $CO_2$ values are determined by this method. Figure S5 is the only place where the MLR and $CO_2$ values are compared – please include in the main manuscript and expand the discussion of the quality of the model results. Figure S5 is a little misleading – it is easy to make it appear that there is good agreement when the model gets the diurnal cycle roughly right. But it does appear that there are large differences hour by hour. It is hard to tell at the scale shown, but it looks like the model is not capturing the morning rush hour peak very well at all.

Pg 6 lines 18-24. How is the intercept of the MLR calculated? Where is the data that shows this? I don't understand how the value of 426 ppm is determined. These MLR

coefficients are key the rest of the interpretation but never clearly explained.

Pg 6 lines 21-24. Why it should be expected that the background is the same in this winter analysis as for the summertime?

Pg 6 lines 32-34. Where do these enhancement percentages come from? I can't see where they are calculated, nor is the $CO_2$ data for these sites ever shown. I understand that the MLR coefficients are useful for interpreting the data, but the $CO_2$ mole fractions need to be shown for each site as well.

Pg 7 lines 3-4. As in a previous comment – how are the correlation uncertainties determined? This is an exciting result but the uncertainties must be shown to be robust to make it believable.

Pg 7 lines 13-18. The two papers cited only discuss NOx and particulates, and I suggest that the authors also refer to the literature on $CO_2$, $CO_2ff$, or CO ratios, for example: Miller, J.B., Lehman, S.J., Montzka, S.A., Sweeney, C., Miller, B.R., Wolak, C., Dlugokencky, E.J., Southon, J.R., Turnbull, J.C., Tans, P.P., 2012. Linking emissions of fossil fuel $CO_2$ and other anthropogenic trace gases using atmospheric $14CO_2$. Journal of Geophysical Research 117, D08302. Turnbull, J.C., Sweeney, C., Karion, A., Newberger, T., Lehman, S.J., Tans, P.P., Davis, K.J., Lauvaux, T., Miles, N.L., Richardson, S.J., Cambaliza, M.O., Shepson, P.B., Gurney, K., Patarasuk, R., Razlivanov, I., 2015. Toward quantification and source sector identification of fossil fuel $CO_2$ emissions from an urban area: Results from the INFLUX experiment. Journal of Geophysical Research: Atmospheres. Lopez, M., Schmidt, M., Delmotte, M., Colomb, A., Gros, V., Janssen, C., Lehman, S.J., Mondelain, D., Perrussel, O., Ramonet, M., Xueref-Remy, I., Bousquet, P., 2013. CO, NOx and $13CO_2$ as tracers for fossil fuel $CO_2$: results from a pilot study in Paris during winter 2010. Atmospheric Chemistry and Physics 13, 7343-7358. Baker, A.K., Beyersdorf, A.J., Doezema, L.A., Katzenstein, A., Meinardi, S., Simpson, I.J., Blake, D.R., Sherwood Rowland, F., 2008. Measurements of nonmethane hydrocarbons in 28 United States cities. Atmospheric Environment 42,

170-182. Warneke, C., McKeen, S.A., de Gouw, J.A., Goldan, P.D., Kuster, W.C., Holloway, J.S., Williams, E.J., Lerner, B.M., Parrish, D.D., Trainer, M., Fehsenfeld, F.C., Kato, S., Atlas, E.L., Baker, A., Blake, D.R., 2007. Determination of urban volatile organic compound emission ratios and comparison with an emissions database. Journal of Geophysical Research 112. Nathan, B., Lauvaux, T., Turnbull, J.C., Gurney, K.R., 2018. Investigations into the use of multi-species measurements for source apportionment of the Indianapolis fossil fuel CO2 signal. Elementa: Science of the Anthropocene 6. Barnes, D.H., 2003. Urban/industrial pollution for the New York City–Washington, D. C., corridor, 1996–1998: 1. Providing independent verification of CO and PCE emissions inventories. Journal of Geophysical Research 108.

Pg 8 lines 1-5. The work in this and previous papers by this group has made huge inroads using low-cost, high-density CO2 sensors to examine urban emissions, and have shown some clear pathways to where this method is useful. Yet the high-quality, low-density systems favoured by other researchers have also provided exciting results. Indeed, this paper uses high-quality measurements to calibrate and validate the low-cost sensor array. It really isn't helpful to pit the two methods against each other as if only one method is valid. Rather, I suggest that the authors reword to emphasize where the methods are complementary.

Pg 8 lines 5-9. "…we do so without the use of computationally intense and heavily parameterized atmospheric transport models." The implication here is that this is the first CO2-based study to achieve this, whereas in fact a number of studies have gone in this direction. See for example the references given above, and: LaFranchi, B.W., Pétron, G., Miller, J.B., Lehman, S.J., Andrews, A.E., Dlugokencky, E.J., Miller, B.R., Montzka, S.A., Hall, B., Neff, W., Sweeney, C., Turnbull, J.C., Wolfe, D.E., Tans, P.P., Gurney, K.R., Guilderson, T.P., 2013. Constraints on emissions of carbon monoxide, methane, and a suite of hydrocarbons in the Colorado Front Range using observations of 14CO2. Atmospheric Chemistry and Physics 13, 11101-11120. Turnbull, J.C., Tans, P.P., Lehman, S.J., Baker, D., Chung, Y., Gregg, J.S., Miller, J.B., Southon, J.R., Zhao,

L., 2011. Atmospheric observations of carbon monoxide and fossil fuel CO2 emissions from East Asia. Journal of Geophysical Research 116. McMeeking, G.R., Bart, M., Chazette, P., Haywood, J.M., Hopkins, J.R., McQuaid, J.B., Morgan, W.T., Raut, J.C., Ryder, C.L., Savage, N., Turnbull, K., Coe, H., 2011. Airborne measurements of trace gases and aerosols over the London metropolitan region. Atmospheric Chemistry and Physics Discussions 11, 30665-30718. Ammoura, L., Xueref-Remy, I., Gros, V., Baudic, A., Bonsang, B., Petit, J.E., Perrussel, O., Bonnaire, N., Sciare, J., Chevallier, F., 2014. Atmospheric measurements of ratios between CO2 and co-emitted species from traffic: a tunnel study in the Paris megacity. Atmospheric Chemistry and Physics 14, 12871-12882.

Pg 8 lines 8-10. The interpretation also requires high quality traffic count information, which is not available everywhere and will be a significant limiting factor.

Figure 7. Suggest showing all data rather than just the median values – figure 5 shows that there is a lot of scatter in the individual measurements that shouldn't be ignored.

Jocelyn Turnbull, GNS Science, June 20, 2018

---

## Referee Comment (RC2) · Anonymous Referee #2 · 18 Jul 2018

General comments:

Shusterman et al. present and analyse results from the low-cost, high-density CO2 monitoring network BEACO2N to demonstrate that such a network allows investigating hyperlocal sources e.g. highway traffic and to track emission changes due to mitigation measures. Key findings, like an experimentally determined correlation length and the strong correlation of local CO2 enhancements with traffic are important results for the urban GHG research community. Overall, the manuscript is very well written, nicely structured and concise. 1.) However, some further detail on the methodology would be instructive for other (and future) researchers attempting to use similar approaches, which would ensure that the paper has the best possible impact. The methods applied are properly referenced, but e.g. the work of de Foy is very recent and some more

information might be useful for the reader. 2.) Furthermore, the authors do not clearly define the terms used for spatial scales e.g. "regionwide" (see specific comments). As different groups/communities use different definitions of "regional" it seems imperative that this is added to the manuscript to avoid confusion. 3.) The authors refer to MRV and that this network would/could be useful. While the work described here echoes the concept of MRV, MRV itself, as introduced by the Bali action plan (UNFCCC), seems not to be the best goal. I would argue that providing atmospheric-based constraints on emissions would be very valuable by itself and can enormously help (local) stakeholders without the complications of being integrated into a MRV framework. After addressing these comments I would fully recommend this work for publication in ACP as it is an important advance in the field and will be of great interest to the community.

Specific comments:

P1L9: consider adding "at subnational scale" as national $CO_2$ emissions are usually fairly easy to report based on consumption data compared to e.g. CFCs, $N_2O$ or $CH_4$ and MRV frameworks exist under UNFCCC (e.g. https://unfccc.int/sites/default/files/non-annex_i_mrv_handbook.pdf). For cities MRV has also been developing (e.g. the GHG protocol), but the authors could highlight the added/complementary value of atmospheric information.

P4 L19: Please give an estimate of what scale "regionwide" refers to.

P4 L25: Why did you choose the 10th percentile to define "regional" and not e.g. the 5th or 20th percentile?

P4 L33: Please correct to "Figure 4"

P5 L3: The daily cycle is mainly driven by boundary layer height dynamics – the local traffic flux is the superimposed fluctuation here, in my opinion. It surely causes a modification e.g. by causing the morning and evening peaks to be more pronounced. However, different studies in rural regions have largely similar diel cycle shapes (e.g. Garcia

et al. 2012 https://www.tandfonline.com/doi/abs/10.3155/1047-3289.58.7.940, Perez et al. 2012, https://www.sciencedirect.com/science/article/pii/S0048969712007498)

P5 L6: It seems counter-intuitive that the PBLH changes earlier in winter (also compared to other studies) as more energy is introduced into the system during summer months to break the NBL (as the solar insulation is stronger and the sun rises earlier). Please provide additional data e.g. PBLH or other atmospheric proxy information in the appendix to support your interpretation.

P5 L32: Why is the other methodology not shown in the supplement and why is this sentence in brackets? Seems to be an interesting finding/information.

P5 L34: You could also refer to the large amount of traffic tunnel studies that have similar findings and are very straightforward (no other source besides traffic) (e.g. references in https://www.atmos-chem-phys.net/14/12871/2014/acp-14-12871-2014.pdf)

P5 L35: One question raised would be how long would you have to observe confirm this 17% trend? Which is answered at P7 L4 for 11-30% emission changes. Consider removing the discussion of the 17% here.

P6 L15: How exactly are the wind speed quartiles subdivided (and why)? See general comment 1.)

P6 L26: Why are Mondays and Saturdays not shown in the supplement?

P6 L29: Could you quantify to which degree the atmospheric dynamics has been controlled for. Claiming that it is "partially controlled for" does not automatically mean that the residual only/primarily reflects emission changes.

P7 L4: What is your confidence of the reported detection of such a trend within 2-3years? 95%? How was this calculated?

P7 L17: The assumption that plumes can be detected within an urban area should be supported e.g. by citations. At scales below 1 km2 it seems that street

canyon effects, building disturbances, etc. could play an important role and hinder the application of concepts such as "plumes", see e.g. Lietzke and Vogt 2013 (https://www.sciencedirect.com/science/article/pii/S1352231013002069) that also investigated traffic emissions at street scale.

P8 L5: I would suggest reconsidering the wording here, especially as you refer to MRV earlier in the manuscript. This work strongly supports the conclusions of Turner et al. 2016, but it seems you have validated and not verified them.

---

## Author Comment (AC1) · 26 Jul 2018

We thank the referee for their comments and time, which have improved our manuscript in many ways, as detailed below:

*This paper uses a set of observations from low-cost, near-surface sensors to examine relationships between $CO_2$ mole fraction and traffic counts, showing strong relationships at individual sites. They also investigate correlation length scales between different sites and develop a multiple linear regression method to establish relationships between $CO_2$ mole fraction and the factors that influence it. The methods described appear to give exciting results showing that traffic fuel efficiency can be monitored by combining these methods with traffic count/flow information.*

*The data and results in this paper are interesting and entirely appropriate for publication in ACP. The major flaw in this paper is that insufficient detail of the methods is given, and mostly only higher level data products (correlation coefficients, MLR coefficients) are given in most places. There are several instances where methods first described in other papers are here described too briefly to be understood without detailed reading of the previous publications. In other cases, figures that are key to understanding the methodology are given only in the supplementary material, and details of the "raw" information that goes into the main figures are lacking. Thus it is difficult to evaluate the robustness of the methods, and readers will have difficulty repeating the analysis or trying it out themselves. ACP doesn't have major length limitations, so the main text should be expanded so that the methodology can be followed. Specific instances are noted in my further comments.*

*The concepts, data, and written language are all good, but I recommend major revisions to expand the explanations of the methodology and show more of the $CO_2$ measurements and comparison to the MLR model. This will allow the reviewers and readers to better evaluate the robustness of the methodology.*

We appreciate the referee's sentiments with regards to both the significance of our results as well as the shortcomings of our manuscript and we endeavor to improve in the ways suggested. In particular, we have added much more detailed explanations of the methodologies based on previous publications for the convenience of the reader. Due to the extremely high volume of $CO_2$ observations involved in a multisite, long-term monitoring campaign, it is not always feasible to present the data in its raw form–hence our preference for "higher level" data products that can summarize large quantities of information relatively succinctly. However, an effort has nonetheless been made to incorporate more of the directly observed $CO_2$ concentrations into the main text wherever possible; case by case explanations of when we did or did not choose to do so are given in response to the specific comments that follow.

*Specific comments:*

*The effect of low-precision measurements is not detailed anywhere in this paper. How does the 0.5 ppm precision impact the results discussed? How is drift in the sensors accounted for, and how will this impact the results, particularly the concept that changes in traffic fuel efficiency could be monitored over time?*

The ±0.5 ppm hourly precision has little impact on the results discussed, given that most of said results are obtained after significant averaging. For example, each dark orange point in Fig. 6 represents, on average, 20 hours, giving each a cumulative error of around ±0.13 ppm, which is invisible on the scale of the graph. When we weight each orange point by its specific uncertainty, we actually find a smaller error (11%) in the slope of the resultant linear regression. We have chosen to report the more conservative, non-weighted slope error (17%) in the text. To underscore the insignificance of the measurement precision, we have revised the text as follows:

"The processed 1-minute averages are assumed to have an uncertainty of less than ±4 ppm, which becomes negligible on the averaging timescales used hereafter."

Any long-term drift in the sensors is accounted for via a combination of periodic (i.e., every 12–24 months) laboratory recalibration and a post hoc data treatment based on an independent reference site in the network domain. This procedure allows us to confidently compare measurements taken multiple years apart, thus enabling inter-annual changes in traffic fuel efficiency to be monitored. The exact details of the calibration and post hoc data treatment are provided in Shusterman et al. (2016) and a full repetition of that discussion is beyond the scope of this manuscript.

*Although the authors acknowledge that traffic contributes only 40% of $CO_2$ emissions, they then focus on only traffic emissions in the analysis. While sites very close to major roads will indeed be strongly influenced by the proximal road, sites further from roads will be influenced by multiple roads as well as other anthropogenic sources, AND by biogenic $CO_2$ sources and sinks. How are these other sources considered? If they are ignored in this analysis, please justify why.*

While traffic contributes "only" 40% of the total $CO_2$ emissions budget for the entire San Francisco Bay Area, this is the single largest source sector in the inventory and the two next largest sectors (industrial/commercial and electricity/co-generation) are often located outside of the urban core studied here. Thus, we safely assume that traffic emissions contribute much more than 40% of the $CO_2$ emissions in the subset of the overall Bay Area where our sensors are located and focus our subsequent analysis on traffic alone.

We acknowledge that other roads do indeed impact the measurements collected at sites located farther from major highways. We do not require that these roads (or other, non-traffic anthropogenic sources) have negligible influence as a premise for our analysis, however we do find our local $CO_2$ enhancements and secondary data products to be well correlated with traffic counts on major highways. While ancillary $CO_2$ sources may account for some of the scatter in our correlations, their insignificance relative to or strong correlation with emissions from major highways is one possible conclusion, rather than a presumption, of our study.

As for the influence of the biosphere, we do not ignore it, but rather adopt data analysis techniques that minimize its importance. Namely, in the multiple linear regression analysis, we separate out the components of the $CO_2$ signal that are correlated with time of year and temperature (likely to be predictive of biosphere activity) from that which is correlated with day of week (unlikely to be predictive of biosphere activity).

*P1 L25: The % of global $CO_2$ emissions from urban areas varies depending on how it is determined. 70 to 80% is probably a better estimate.*

We thank the referee for providing a more accurate range and have updated the text appropriately:

"Currently, an estimated 70–80% of global $CO_2$ emissions are urban in origin and this fraction is expected to grow as migration to urban areas continues and intensifies with the industrialization of developing nations (United Nations, 2011)."

*Section 2.2. Traffic counts. For the sites very close to a particular highway, traffic monitor data for that nearby highway makes sense. For sites that are further from any particular highway, even if traffic is the dominant proximal $CO_2$ source, surely more than one highway (and local roads as well) will contribute to the signal observed at that site. How are multiple sources accounted for?*

We agree that accounting only for the influence of a single highway (rather than multiple highways and/or additional local roads) is a first order approximation of the total traffic emissions influencing a given $CO_2$ monitor, especially those situated at greater distances from said highway. Interestingly, we nonetheless find this first order approach to produce robust correlations with the observed fluctuations in $CO_2$. As previously stated in our response to an earlier comment, the predominance of these single highway emissions (and/or their strong correlation with those from other sources) is a possible conclusion, rather than a presumption, of our study.

*P3 L31–32; P4 L1–3: Please expand to explain the methodology used here in enough detail to be followed without requiring the reader to refer to the McKain paper. Are these correlation lengths determined using $CO_2$ mole fraction, or the enhancements in $CO_2$ relative to the background? Where is the raw data that is used to derive these correlation lengths? Plots of the $CO_2$ time series should be included (these could go in the supplementary material).*

We have expanded the manuscript text to include a more thorough explanation of the methodology used here:

"To quantify the spatial heterogeneity present across the network, we examine the degree of correlation between every possible pairing of sites in a given season as a function of the distance between them, borrowing from a similar analysis used by McKain et al. (2012). For straightforward comparison with the McKain et al. results, we first average the total $CO_2$ mole fractions to 5-minute resolution. Then, for every pairwise combination of two sites, we perform an ordinary least squares linear regression between the two 5-minute time series and calculate the Pearson correlation coefficient. We repeat this procedure after offsetting the two time series by ±5 minutes, ±10 minutes, etc., allowing for up to a ±3-h lag and choose the optimal $r^2$ value from the

possible offsets. We plot the thus optimized pairwise correlations as a function of the distance separating the two relevant sites (Figs. 2 and 3) and fit the results to a single term exponential decay on top of a constant background, defined by the mean correlation observed at inter-site distances greater than 20 km."

Given the sheer volume of data used to derive these correlation lengths (3–7 months of data at 5-minute resolution from 28 different sites), we elect to not include $CO_2$ time series in the supplementary material, as such plots are too visually cramped to be interpreted easily. Instead, we refer interested readers to the Data Availability statement, where we have included a public link to all $CO_2$ datasets used in this manuscript.

*P4 L6–14: Please explain what the correlation lengths should be interpreted to mean. I take it that a shorter correlation length implies more influence of sources close to the sites. Longer length scales would imply more influence of sources further away? The referenced studies are all about pollutant gases, not $CO_2$–it might be reasonable to expect higher correlations and longer length scales for a long-lived gas like $CO_2$ with large and varying background.*

The referee has correctly inferred our intended interpretation of the correlation lengths, and we have updated the text appropriately to make this explicit:

"The characteristic length scale of this correlation is 2.9 km (defined as the e-folding distance of the exponential fits in Fig. 2; 3.6 km during the day and 2.2 km at night), which we interpret as an indicator of the distance at which various emission sources exert influence over a site's measurements. Shorter correlation lengths indicate sensitivity to near-field emissions, while longer correlation lengths imply sensitivity to far-field phenomena."

We have also edited the discussion of the prior studies of reactive pollutant gases according to the referee's comments:

"In either season, the correlation lengths are, as expected, considerably longer than the previously observed ~100 to 1000-m e-folding distances of reactive urban pollutants that are also lost via chemical pathways (e.g., Zhu et al., 2006; Beckerman et al., 2008; Choi et al., 2014), thus validating the original choice of 2 km as the desirable inter-site separation in the design of the BEACO$_2$N instrument."

*P4 L18–19: I don't follow why the daytime correlations imply this information. Please clarify.*

We have edited the text and added a figure to clarify our intended interpretation of the daytime correlations:

"However, McKain et al. saw very little correlation after restricting their analysis to daytime hours, even at very short (<5 km) inter-site distances, which implies that daytime observations reflect hyperlocal phenomena only. In contrast, we observe moderate to high correlations during the day, which illustrates that information about emissions and transport phenomena on a variety of scales is preserved. A spatial visualization of the daytime correlation coefficients at four representative winter sites is shown in Fig. 4. We see that PER is well correlated with its neighbors only,

suggesting the presence of local phenomena that do not affect other parts of the network. LCC, however, also exhibits relationships with more distant sites, indicating a sensitivity to more regional-scale (10–30 km) influences. Meanwhile, HRS and OHS each possess at least one near neighbor with whom they are poorly correlated, perhaps due to hyperlocal events specific to those sites. While the region-wide phenomena can be characterized using sparser networks of high-cost, conventional monitoring equipment, the ability to capture these local processes is unique to the high-density approach."

*P4 L33: Figure 4, not Fig. 2?*

This typographical error has been corrected.

*P5 L1–10: It is curious that the amplitude of the diurnal cycle is larger in winter than in summer. The timing of the diurnal pattern shown in Fig. 4 doesn't quite gel with the argument that this is due to the lower/stronger daytime boundary layers. Have the authors considered the influence of biogenic $CO_2$ fluxes, which may be more important in the rainy winter season in San Francisco than in the summer?*

The precise diurnal cycle and relative strength of the summertime vs. wintertime daytime boundary layers in the San Francisco Bay Area are not well understood, so we find this potential explanation of the $CO_2$ diurnal cycles to be nonetheless plausible if not precisely correct. We do, however, agree that the influence of biogenic $CO_2$ fluxes may be an important alternative or additional consideration, and have updated the text accordingly:

"This diurnal profile corresponds well with known patterns in traffic emissions–which are largely consistent across seasons–superimposed on diel fluctuations in boundary layer height and/or biosphere activity that vary in timing and magnitude according to the season. Namely, these results might be interpreted to conclude the nighttime boundary layer in the BEACO$_2$N domain to be shallower during the winter months, producing a larger regional increase in response to rush hour traffic. The wintertime layer also appears to expand and re-contract earlier in the day than the summertime layer, resulting in both an earlier minimum and an earlier rise in afternoon–evening concentrations. The larger amplitude of the wintertime diurnal cycle may also reflect the greater influence of daytime photosynthesis and nighttime respiration during the San Francisco Bay Area's rainy winter season."

*P5 L 26–35: I pity those commuters who are contributing to high traffic flows at 4 am!*

Agreed!

*"An alternative analysis using traffic density…" I think I understand that this is an attempt to examine how congested vs free-flowing traffic might change the results? Please clarify.*

*"We observe a factor of 2 difference in the local $CO_2$ between congested vs. free-flowing conditions"–which is higher?*

The referee has interpreted our reference to traffic density correctly; we subsequently observed congestion to lead to higher local $CO_2$ enhancements. The text has been updated to provide a clarification with regards to these two comments:

"An alternative analysis using traffic density–obtained by dividing the traffic flow by the average vehicle speed–yields almost identical results (Fig. S5), revealing a factor of 2 increase in the local $CO_2$ during congestion (high traffic flow/density) relative to free-flowing conditions (low traffic flow/density), similar to that observed by a previous on-road mobile monitoring study by Maness et al. (2015)."

*How is the regression slope determined, and how is the uncertainty determined? This needs further detail, particularly because the regression slope is determined not from the full dataset, but from a fit to the median values. The idea that trends in fuel efficiency can be tracked by this method is tantalizing, but the statistics must be demonstrated to be robust.*

An explanation of our regression methodology has been added to the text:

"Also shown in Fig. 6 are the median $CO_2$ concentrations observed in each 500 veh h$^{-1}$ traffic flow increment and the ordinary least squares linear regression through these binned medians."

"The standard error of the slope of the linear regression is calculated as the standard deviation of the model–data $CO_2$ residuals divided by the square root of the sum of the squared differences between each traffic flow increment and the mean traffic flow."

*P6 L5–17: Not enough information is given to understand how these MLRs are constructed and therefore how they can be interpreted. Please expand on the method and provide further details on which factors were most important. On the following page (lines 5–12), there is discussion about how improved resolution of the meteorological datasets would help, but nowhere is the current resolution and limitations of the data explained!*

*It isn't clear how the "modelled" $CO_2$ values are determined by this method. Figure S5 is the only place where the MLR and $CO_2$ values are compared–please include in the main manuscript and expand the discussion of the quality of the model results. Figure S5 is a little misleading–it is easy to make it appear that there is good agreement when the model gets the diurnal cycle roughly right. But it does appear that there are large differences hour by hour. It is hard to tell at the scale shown, but it looks like the model is not capturing the morning rush hour peak very well at all.*

We have expanded what was formerly Fig. S5 to include the other four sites and moved it to the main text. In updating Fig. S5 (now Fig. 7), an attempt has also been made to allow for closer examination of model–observation agreement on short timescales by depicting a representative week (rather than an entire month) of data. We have also added a table describing the relative importance of the various MLR factors and updated the text to include a more detailed description of the MLR analysis, the limitations of the meteorological datasets, as well as an explanation and discussion of the model–observation comparison, as follows:

"Briefly, we use an ordinary least squares linear regression to calculate the best fit of the relationship between a site's $CO_2$ signal and temperature, specific humidity, wind, boundary layer height, time of day, day of week, and time of year. Hourly measurements of temperature, specific humidity, wind speed, and wind direction are taken from a single NOAA Integrated Surface Database weather station at the Port of Oakland International Airport (https://www.ncdc.noaa.gov/isd/) and 3-hour boundary layer heights are provided at $0.125^\circ$ by $0.125^\circ$ resolution by the ECMWF's ERA-Interim model (Dee et al., 2011; http://apps.ecmwf.int/datasets/). Although the low spatio-temporal resolution of these datasets limits their ability to capture hyperlocal meteorologies, here we follow the example of de Foy, who was nonetheless able to derive meaningful results from similarly coarse weather products.

The nonlinear relationship between $CO_2$ concentrations and wind or boundary layer height is captured by dividing these meteorological datasets into quartiles and assigning each observation a value between 0 (at the maximum of the quartile) and 1 (at the minimum) using piecewise linear interpolation. The wind speed quartiles are further subdivided by wind direction and reassigned values of 0–1 accordingly before fitting a linear coefficient to each subset. The time of year is represented as a sum of sines and cosines with annual or semiannual periodicities whose values also vary between 0 and 1 and whose amplitudes are determined by the linear regression. Zeroes and ones are used to designate each hour of each type of day of the week as well. For example, timesteps corresponding to 0800 LT on a Monday may be assigned a 1 while all other timesteps are set to zero before the linear regression is performed. As a result, the MLR factors derived for each of the preceding explanatory variables can be interpreted in units of ppm $CO_2$. Meanwhile, the temperature and specific humidity variables are treated by calculating their difference from their mean values and dividing by their respective standard deviations before each is fit to $CO_2$ with a single linear coefficient, which will have units of ppm $^\circ C^{-1}$ and ppm $(kg_{water}\ kg_{air}^{-1})^{-1}$, respectively.

The independent variable leading to the greatest square of the Pearson correlation coefficient is then combined with each of the remaining variables and a second regression is performed. The two-input combination leading to the largest increase in the correlation coefficient is then combined with each of the remaining variables, and so on, until the addition of a new independent variable no longer increases the $r^2$ value by at least 0.005.

For this analysis, we use hourly total $CO_2$ concentrations (the sum of the local and regional components) measured at five sites between 15 February 2017 and 15 February 2018. For each site, the optimal set of explanatory variables and their relative contributions to the correlation coefficient are given in Table 2. Summing the products of each of the MLR factors with their respective independent variables (e.g., time of day, wind speed, etc.) gives the mixing ratio predicted by the MLR model; a representative week of observed and modeled $CO_2$ concentrations is shown in Fig. 7. We find generally good agreement, with some significant hour-by-hour model–observation differences, especially at RFS. These do not, however, appear to be systematic either in sign or in timing (e.g., the rush hour peak in $CO_2$ may be poorly modeled on one day but well predicted on another)."

*P6 L18–24: How is the intercept of the MLR calculated? Where is the data that shows this? I don't understand how the value of 426 ppm is determined. These MLR coefficients are key to the rest of the interpretation but never clearly explained.*

The intercept of the MLR analysis is defined as the modeled $CO_2$ concentration when all of the input variables possess a value of zero. More details of the MLR analysis have been added in response to previous comments.

*P6 L21–24: Why should it be expected that the background is the same in this winter analysis as for the summertime?*

The intercept calculated using the MLR analysis is not a wintertime intercept, but rather an annual average intercept, as the MLR analysis spans 15 February 2017 to 15 February 2018. While we do not expect the annual average background to agree perfectly with the summertime background concentration, we find it nonetheless interesting to note that the annual value agrees more closely with the wintertime value than the summertime one.

*P6 L32–34: Where do these enhancement percentages come from? I can't see where they are calculated, nor is the $CO_2$ data for these sites ever shown. I understand that the MLR coefficients are useful for interpreting the data, but the $CO_2$ mole fractions need to be shown for each site as well.*

The weekday enhancements are the maximum difference between Tuesday–Thursday and Sunday hourly MLR factors, expressed as a percentage of the Sunday (lower) factor. We have updated the text appropriately to clarify this point, and have also added a figure depicting the $CO_2$ mole fractions from which the MLR factors are derived:

"The dependencies on time of day and day of week derived via this method are hypothesized to primarily reflect the changes in emissions, as the influence of the coincident changes in atmospheric dynamics has been at least partially controlled for. For reference, the corresponding Tuesday–Thursday and Sunday diel cycles in the total $CO_2$ observed at each site are shown in Fig. 9. Indeed, we do observe some of the same intuitive patterns in the linear regression coefficients, such as higher coefficients on weekday mornings corresponding to higher rush hour traffic emissions on those days, but with greater opportunity to differentiate between days of the week, especially around noon when raw concentrations are generally similar. As expected, the Tuesday–Thursday enhancement in the MLR factors is larger at the sites located close to a freeway (e.g., up to 520% of the corresponding Sunday MLR factor at FTK) but is less pronounced at LBL (70%), which is farther away from major mobile sources."

*P7 L3–4: As in a previous comment–how are the correlation uncertainties determined? This is an exciting result but the uncertainties must be shown to be robust to make it believable.*

Please refer to our response to the previous comment concerning correlation uncertainties. We have also added a discussion of the relevant confidence intervals to the text:

"Assuming a steady decrease in emissions of 3.5% per year, one BEACO₂N site is therefore sufficiently sensitive to detect such a trend with 68% confidence in as little as 3 years. By leveraging multiple independent sites, even greater confidence and/or shorter timescales could be achieved."

*P7 L13–18: The two papers cited only discuss $NO_x$ and particulates, and I suggest that the authors also refer to the literature on $CO_2$, $CO_2ff$, or CO ratios, for example:*

*Miller, J. B., Lehman, S. J., Montzka, S. A., Sweeney, C., Miller, B. R., Wolak, C., Dlugokencky, E. J., Southon, J. R., Turnbull, J. C., and Tans, P. P.: Linking emissions of fossil fuel $CO_2$ and other anthropogenic trace gases using atmospheric $^{14}CO_2$, J. Geophys. Res., 117, D08302, 2012.*

*Turnbull, J. C., Sweeney, C., Karion, A., Newberger, T., Lehman, S. J., Tans, P. P., Davis, K. J., Lauvaux, T., Miles, N. L., Richardson, S. J., Cambaliza, M. O., Shepson, P. B., Gurney, K., Patarasuk, R., and Razlivanov, I.: Toward quantification and source sector identification of fossil fuel $CO_2$ emissions from an urban area: Results from the INFLUX experiment, J. Geophys. Res. Atmos., 120, 292–312, 2015.*

*Lopez, M., Schmidt, M., Delmotte, M., Colomb, A., Gros, V., Janssen, C., Lehman, S. J., Mondelain, D., Perrussel, O., Ramonet, M., Xueref-Remy, I., and Bousquet, P.: CO, $NO_x$ and $^{13}CO_2$ as tracers for fossil fuel $CO_2$: results from a pilot study in Paris during winter 2010, Atmos. Chem. Phys., 13, 7343–7358, 2013.*

*Baker, A. K., Beyersdorf, A. J., Doezema, L. A., Katzenstein, A., Meinardi, S., Simpson, I. J., Blake, D. R., and Sherwood Rowland, F.: Measurements of nonmethane hydrocarbons in 28 United States cities, Atmos. Environ., 42, 170-182, 2008.*

*Warneke, C., McKeen, S. A., de Gouw, J. A., Goldan, P. D., Kuster, W. C., Holloway, J. S., Williams, E. J., Lerner, B. M., Parrish, D. D., Trainer, M., Fehsenfeld, F. C., Kato, S., Atlas, E. L., Baker, A., and Blake, D. R.: Determination of urban volatile organic compound emission ratios and comparison with an emissions database, J. Geophys. Res., 112, D10S46, 2007.*

*Nathan, B., Lauvaux, T., Turnbull, J. C., and Gurney, K. R.: Investigations into the use of multi-species measurements for source apportionment of the Indianapolis fossil fuel $CO_2$ signal, Elem. Sci. Anth., 6, 2018.*

*Barnes, D. H., Wofsy, S. C., Fehlau, B. P., Gottlieb, E. W., Elkins, J. W., Dutton, G. S., and Montzka, S. A.: Urban/industrial pollution for the New York City–Washington, D. C., corridor, 1996–1998: 1. Providing independent verification of CO and PCE emissions inventories, J. Geophys. Res., 108, 4185, 2003.*

We thank the referee for providing such an extensive list of possible references; the Lopez et al. (2013), Nathan et al. (2018), and Turnbull et al. (2015) studies were found to be most relevant to the discussion and have been added to the text.

*P8 L1–5: The work in this and previous papers by this group has made huge inroads using low-cost, high-density $CO_2$ sensors to examine urban emissions, and have shown some clear pathways to where this method is useful. Yet the high-quality, low-density systems favoured by other researchers have also provided exciting results. Indeed, this paper uses high-quality measurements to calibrate and validate the low-cost sensor array. It really isn't helpful to pit the two methods against each other as if only one method is valid. Rather, I suggest that the authors reword to emphasize where the methods are complementary.*

We appreciate and agree with this perspective on the complementarity of the two approaches and have revised the text accordingly:

"Most prior studies of urban $CO_2$ emissions (e.g., McKain et al., 2012; Kort et al., 2013; Wu et al., 2018) have favored sparser networks of high-quality instruments, finding this approach to be better

suited for resolving small trends in total region-wide emissions. It is hypothesized that the ideal monitoring strategy depends on the particular goals and location of a given application, with certain locales and emission sources necessitating high-cost, low-density instrumentation, complemented by other domains where low-cost, high-density platforms are more effective. The potential trade-offs between measurement quality and instrument quantity specific to the San Francisco Bay Area have been investigated previously using an ensemble of observing system simulations by Turner et al. (2016), who found BEACO$_2$N-like observing systems to outperform smaller, higher quality networks in estimating regional as well as more localized emission phenomena there. While Turner et al. saw significant benefits to achieving an instrument precision of 1 ppm, further increases in measurement quality offered little advantage in constraining emissions, especially those from line and point sources."

*P8 L5–9: ". . .we do so without the use of computationally intense and heavily parameterized atmospheric transport models." The implication here is that this is the first CO$_2$-based study to achieve this, whereas in fact a number of studies have gone in this direction. See for example the references given above, and:*
*LaFranchi, B. W., Pétron, G., Miller, J. B., Lehman, S. J., Andrews, A. E., Dlugokencky, E. J., Miller, B. R., Montzka, S. A., Hall, B., Neff, W., Sweeney, C., Turnbull, J. C., Wolfe, D. E., Tans, P. P., Gurney, K. R., and Guilderson, T. P.: Constraints on emissions of carbon monoxide, methane, and a suite of hydrocarbons in the Colorado Front Range using observations of $^{14}$CO$_2$, Atmos. Chem. Phys., 13, 11101–11120, 2013.*
*Turnbull, J. C., Tans, P. P., Lehman, S. J., Baker, D., Chung, Y., Gregg, J. S., Miller, J. B., Southon, J. R., and Zhao, L.: Atmospheric observations of carbon monoxide and fossil fuel CO$_2$ emissions from East Asia, J. Geophys. Res., 116, D24306, 2011.*
*McMeeking, G. R., Bart, M., Chazette, P., Haywood, J. M., Hopkins, J. R., McQuaid, J. B., Morgan, W. T., Raut, J. C., Ryder, C. L., Savage, N., Turnbull, K., and Coe, H.: Airborne measurements of trace gases and aerosols over the London metropolitan region. Atmos. Chem. Phys, 11, 5163–5187, 2011.*
*Ammoura, L., Xueref-Remy, I., Gros, V., Baudic, A., Bonsang, B., Petit, J. E., Perrussel, O., Bonnaire, N., Sciare, J., Chevallier, F.: Atmospheric measurements of ratios between CO$_2$ and co-emitted species from traffic: a tunnel study in the Paris megacity, Atmos. Chem. Phys., 14, 12871–12882, 2014.*

We do not intend to imply that we are the first CO$_2$-based study to employ simpler models to constrain emissions; we appreciate and acknowledge the important prior work in this vein referenced by the referee. Our intention was rather to contextualize this work relative to the Turner et al. (2016) study referenced in the previous sentence, in which it is suggested that low-cost monitoring frameworks can be fruitfully leveraged using computationally intense atmospheric transport models. The text is simply meant to communicate that we agree low-cost frameworks can be fruitfully leveraged, even without said transport models.

*P8 L8–10: The interpretation also requires high quality traffic count information, which is not available everywhere and will be a significant limiting factor.*

We agree with the referee that the particular analysis detailed in this manuscript relies on the availability of traffic count information, although we do note that such information does exist for

many metropolitan areas (e.g., http://pems.dot.ca.gov/, http://transportation.austintexas.io/radar/) and ongoing efforts to make these datasets public are progressing in both the academic (https://www.cattlab.umd.edu/) and private (http://www.traffic.com/) sectors. We have, however, added a disclaimer to the manuscript text:

"Furthermore, we show that a multiple linear regression analysis allows the signature of highway traffic to be extracted from sites located throughout the network, enabling trends in mobile emissions to be quantified without specially situated, roadside monitors. Although this approach requires real time traffic count information that is not yet available in all locations, our finding is nonetheless an important result, as deriving and implementing a particular, a priori network layout is a non-trivial task."

*FIG7: I suggest showing all data rather than just the median values–Fig. 5 shows that there is a lot of scatter in the individual measurements that shouldn't be ignored.*

We believe that a typographical error in the caption for Fig. 7 (now Fig. 10) has led to a misunderstanding about the data that is depicted in said figure. The points shown on these plots are not median values, as the caption originally stated, but rather the entirety of the dataset of MLR coefficients derived to reflect the dependence of the $CO_2$ concentrations upon time of day and day of week during morning hours (0400–0800 LT); thus there exists no "scatter" to be shown. The caption has been corrected to clarify this point:

"Morning (0400–0800 LT) multiple linear regression coefficients shown as a function of summertime traffic flow; black solid lines indicate the linear regression through the MLR factors (equations given above each subplot) and gray dashed lines show the uncertainty in the regression slope."

*References:*

Shusterman, A. A., Teige, V. E., Turner, A. J., Newman, C., Kim, J., and Cohen, R. C.: The BErkeley Atmospheric $CO_2$ Observation Network: initial evaluation, Atmos. Chem. Phys., 16, 13449–13463, doi:10.5194/acp-16-13449-2016, 2016.

Turner, A. J., Shusterman, A. A., McDonald, B. C., Teige, V., Harley, R. A., and Cohen. R. C.: Network design for quantifying urban $CO_2$ emissions: assessing trade-offs between precision and network density, Atmos. Chem. Phys., 16, 13465–13475, doi:10.5194/acp-16-13465-2016, 2016.

---

## Author Comment (AC2) · 26 Jul 2018

We appreciate the referee's time and feedback, which have resulted in significant improvements to our manuscript, as detailed below:

*Shusterman et al. present and analyze results from the low-cost, high-density $CO_2$ monitoring network BEACO$_2$N to demonstrate that such a network allows investigating hyperlocal sources, e.g., highway traffic, and to track emission changes due to mitigation measures. Key findings, like an experimentally determined correlation length and the strong correlation of local $CO_2$ enhancements with traffic are important results for the urban GHG research community. Overall, the manuscript is very well written, nicely structured, and concise.*

*1.) However, some further detail on the methodology would be instructive for other (and future) researchers attempting to use similar approaches, which would ensure that the paper has the best possible impact. The methods applied are properly referenced, but, e.g., the work of de Foy is very recent and some more information might be useful for the reader.*

The referee's request for a greater level of detail in our methodological descriptions is a sentiment shared by the other referee as well. We have adjusted the text accordingly, as detailed in response to the specific comments below as well as our other referee response.

*2.) Furthermore, the authors do not clearly define the terms used for spatial scales, e.g., "regionwide" (see specific comments). As different groups/communities use different definitions of "regional," it seems imperative that this is added to the manuscript to avoid confusion.*

Please see our responses to the specific comments below.

*3.) The authors refer to MRV and that this network would/could be useful. While the work described here echoes the concept of MRV, MRV itself, as introduced by the Bali action plan (UNFCCC), seems not to be the best goal. I would argue that providing atmospheric-based constraints on emissions would be very valuable by itself and can enormously help (local) stakeholders without the complications of being integrated into an MRV framework.*

We acknowledge that some readers, the referee included, may adhere to a much stricter definition of MRV activities and have removed all references to MRV from the text in favor of language referring to atmospheric-based constraints on emissions more generally; see our responses to the specific comments below for details.

*After addressing these comments, I would fully recommend this work for publication in ACP as it is an important advance in the field and will be of great interest to the community.*

*Specific comments:*

*P1 L9: Consider adding "at subnational scale" as national $CO_2$ emissions are usually fairly easy to report based on consumption data compared to, e.g., CFCs, $N_2O$, or $CH_4$, and MRV frameworks exist under UNFCCC (e.g., https://unfccc.int/sites/default/files/non-annex_i_mrv_handbook.pdf). For cities MRV has also been developing (e.g., the GHG protocol), but the authors could highlight the added/complementary value of atmospheric information.*

We have updated the text to refer to subnational scales and have removed this and all subsequent references to MRV frameworks:

"Urban carbon dioxide comprises the largest fraction of anthropogenic greenhouse gas emissions and yet quantifying urban emissions at subnational scales is highly challenging, as numerous emission sources reside in close proximity within each topographically intricate urban dome."

"To support this effort, there is a clear need for monitoring strategies capable of describing emission changes and attributing those changes to the relevant policy measures (Pacala et al., 2010)."

"However, cities also present the largest atmospheric monitoring challenge in that many disparate emission sources combine with complex topography."

"A considerable amount of emission estimation work has been invested in the development of activity-based emission inventories for selected metropolitan areas […]"

*P4 L19: Please give an estimate of what scale "regionwide" refers to.*

While the specific sentence to which the referee refers no longer exists in its original form, we have updated the first reference to regional spatial scales in the text to clarify our intended meaning of the term:

"LCC, however, also exhibits relationships with more distant sites, indicating a sensitivity to more regional-scale (10–30 km) influences."

*P4 L25: Why did you choose the 10th percentile to define "regional" and not, e.g., the 5th or 20th percentile?*

As mentioned in the text, the 10th percentile is chosen "to account for measurement error […] as well as any nearfield draw down from the local biosphere." We note in Sect. 2.1 that Shusterman et al. (2016) found the 1-min mean measurements from the $BEACO_2N$ $CO_2$ monitors to possess an uncertainty of less than ±4 ppm, which amounts to between 5% and 10% of the typical ambient $CO_2$ signals observed in our urban domain. We therefore adopt the conservative upper limit of 10% to allow for some influence from the biosphere, although a precise quantification of this component of the signal is beyond the scope of this study. Thus, assuming an overall 10% uncertainty in an arbitrarily chosen site's ability to characterize the regional signal, we define the bottom 10th percentile of the observations as our best estimate of this quantity.

The manuscript text has been updated to direct the reader to the reasoning behind this quantity:

"The bottom 10th percentile is chosen (rather than the absolute minimum) to account for measurement error (±4 ppm at 1-min resolution; see Shusterman et al., 2016) as well as any nearfield draw down from the local biosphere; negative values in the local signals are likely attributable to some combination of these effects."

*P4 L33: Please correct to "Figure 4."*

This typographical error has been corrected.

*P5 L3: The daily cycle is mainly driven by boundary layer height dynamics–the local traffic flux is the superimposed fluctuation here, in my opinion. It surely causes a modification, e.g., by causing the morning and evening peaks to be more pronounced. However, different studies in rural regions have largely similar diel cycle shapes (e.g., Garcia et al., 2012 https://www.tandfonline.com/doi/abs/10.3155/1047-3289.58.7.940; Perez et al., 2012 https://www.sciencedirect.com/science/article/pii/S0048969712007498).*

We agree with the referee that the traffic flux is the superimposed fluctuation here, and the text as written reflects this sentiment:

"This diurnal profile corresponds well with known patterns in traffic emissions–which are largely consistent across seasons–superimposed on diel fluctuations in boundary layer height and/or biosphere activity that vary in timing and magnitude according to the season."

*P5 L6: It seems counter-intuitive that the PBLH changes earlier in winter (also compared to other studies), as more energy is introduced into the system during summer months to break the NBL (as the solar insulation is stronger and the sun rises earlier). Please provide additional data, e.g., PBLH or other atmospheric proxy information in the appendix to support your interpretation.*

Unfortunately, there exist no direct PBLH observations in the area with adequate temporal resolution to inform this analysis. Instead we show the median diel cycles in the summer vs. wintertime temperatures and wind speeds observed at the Port of Oakland International Airport's NOAA Integrated Surface Database station (https://www.ncdc.noaa.gov/isd/) below:

[Figure]

We see that the increases in atmospheric proxies that might be associated with PBLH changes occur at almost identical times of day across seasons, even if the sun rises earlier and more energy is introduced into the system overall during the summer months, as the referee suggests. We do acknowledge, however, that the seasonal differences in PBLH changes are not the only possible explanation for the difference in the diel cycle in regional $CO_2$ concentrations, and have updated the text to reflect an additional possibility suggested by the other referee:

"Namely, these results might be interpreted to conclude the nighttime boundary layer in the BEACO2N domain to be shallower during the winter months, producing a larger regional increase in response to rush hour traffic. The wintertime layer also appears to expand and re-contract earlier in the day than the summertime layer, resulting in both an earlier minimum and an earlier rise in afternoon–evening concentrations. The larger amplitude of the wintertime diurnal cycle may also reflect the greater influence of daytime photosynthesis and nighttime respiration during the San Francisco Bay Area's rainy winter season."

*P5 L32: Why is the other methodology not shown in the supplement and why is this sentence in brackets? Seems to be an interesting finding/information.*

We have removed the parentheses around this statement and have added a figure to the supplement that illustrates the results of this alternative methodology.

*P5 L34: You could also refer to the large amount of traffic tunnel studies that have similar findings and are very straightforward (no other source besides traffic), e.g., references in https://www.atmos-chem-phys.net/14/12871/2014/acp-14-12871-2014.pdf.*

We appreciate the referee's suggestions of additional related studies and believe that such tunnel-based measurement campaigns contribute very important information to mobile emission estimation efforts. However, in the interest of succinctness, we choose to forego a broader discussion of the many analyses that use $CO_2$ as a baseline against which the concentrations of co-emitted species are normalized and instead limit our discussion to studies that analyze the traffic dependence of $CO_2$ concentrations in their own right (i.e., Maness et al., 2015).

*P5 L35: One question raised would be how long would you have to observe to confirm this 17% trend? Which is answered at P7 L4 for 11–30% emission changes. Consider removing the discussion of the 17% here.*

As suggested, we have moved this discussion to occur later in the manuscript.

*P6 L15: How exactly are the wind speed quartiles subdivided (and why)? See general comment 1.)*

As noted in the text, the wind speed quartiles are subdivided to allow for a "nonlinear relationship" between $CO_2$ concentrations and this explanatory variable. In Gaussian dispersion modeling, for example, the downwind concentration of a given pollutant is inversely (rather than linearly) proportional to wind speed. Because our regression method is by definition linear, subdividing the wind speeds in this way allows us to decompose more complex mathematical relationships into

linear components. We have updated the text to give more detail regarding the exact methodology of this approach:

"The nonlinear relationship between $CO_2$ concentrations and wind or boundary layer height is captured by dividing these meteorological datasets into quartiles and assigning each observation a value between 0 (at the maximum of the quartile) and 1 (at the minimum) using piecewise linear interpolation. The wind speed quartiles are further subdivided by wind direction and reassigned values of 0–1 accordingly before fitting a linear coefficient to each subset. The time of year is represented as a sum of sines and cosines with annual or semiannual periodicities whose values also vary between 0 and 1 and whose amplitudes are determined by the linear regression. Zeroes and ones are used to designate each hour of each type of day of the week as well. For example, timesteps corresponding to 0800 LT on a Monday may be assigned a 1 while all other timesteps are set to zero before the linear regression is performed. As a result, the MLR factors derived for each of the preceding explanatory variables can be interpreted in units of ppm $CO_2$. Meanwhile, the temperature and specific humidity variables are treated by calculating their difference from their mean values and dividing by their respective standard deviations before each is fit to $CO_2$ with a single linear coefficient, which will have units of ppm $^oC^{-1}$ and ppm $(kg_{water} \ kg_{air}^{-1})^{-1}$, respectively.

The independent variable leading to the greatest square of the Pearson correlation coefficient is then combined with each of the remaining variables and a second regression is performed. The two-input combination leading to the largest increase in the correlation coefficient is then combined with each of the remaining variables, and so on, until the addition of a new independent variable no longer increases the $r^2$ value by at least 0.005."

*P6 L26: Why are Mondays and Saturdays not shown in the supplement?*

A figure depicting MLR factors derived for Mondays, Fridays, and Saturdays has been added to the supplement.

*P6 L29: Could you quantify to which degree the atmospheric dynamics have been controlled for? Claiming that it is "partially controlled for" does not automatically mean that the residual only/primarily reflects emission changes.*

Without knowledge of the true emissions within a given site's footprint of sensitivity, we cannot quantify the degree to which atmospheric dynamics have been controlled for. The fact that the MLR factors remaining after "partially" controlling for dynamics may primarily reflect emission changes is a hypothesis rather than a premise of this study, a hypothesis that the discussion goes on to support with a first order, proof-of-concept analysis of the diel cycles in these factors. We have updated the text to clarify the speculative nature of this claim, and also to provide additional detail regarding the diel cycle analysis:

"The dependencies on time of day and day of week derived via this method are hypothesized to primarily reflect the changes in emissions, as the influence of the coincident changes in atmospheric dynamics has been at least partially controlled for. For reference, the corresponding Tuesday–Thursday and Sunday diel cycles in the total $CO_2$ observed at each site are shown in Fig. 9. Indeed, we do observe some of the same intuitive patterns in the linear regression coefficients,

such as higher coefficients on weekday mornings corresponding to higher rush hour traffic emissions on those days, but with greater opportunity to differentiate between days of the week, especially around noon when raw concentrations are generally similar. As expected, the Tuesday–Thursday enhancement in the MLR factors is larger at the sites located close to a freeway (e.g., up to 520% of the corresponding Sunday MLR factor at FTK) but is less pronounced at LBL (70%), which is farther away from major mobile sources."

*P7 L4: What is your confidence of the reported detection of such a trend within 2–3 years? 95%? How was this calculated?*

The stated uncertainty in the regression slopes (11–30%) is the standard error, i.e., the 68% confidence interval. Assuming that the 35% reduction in $CO_2$ emissions per vehicle required by fuel efficiency regulation occurs evenly over ~10 years necessitates a 3.5% change in $CO_2$ emissions per vehicle per year. Thus, with a regression uncertainty of 11%, this 3.5% annual trend is detectable within just over 3 years using the observations from a single site. Even modest improvements in our ability to leverage information from $N > 1$ sites within the network would allow for trend detection with greater confidence and/or shorter timescales if, for example, different sites' observations are found to be sufficiently independent to scale down the uncertainty by $\sqrt{N}$. We have updated the manuscript text to clarify this point, include the re-located discussion of the LAN 17% slope uncertainty, and present the timescale of detection more precisely:

"When we examine the relationship between these multiple linear regression coefficients and morning traffic flow as we did at LAN (Fig. 10), we again find positive correlations. The standard error of the slope of the linear regression is calculated as the standard deviation of the model–data $CO_2$ residuals divided by the square root of the sum of the squared differences between each traffic flow increment and the mean traffic flow. The uncertainty in the slopes is thus found to be 11–30%, indicating that analysis of a single site could be used to detect as small as 11% changes in average emissions per vehicle, an improvement upon the 17% slope uncertainty calculated for the near-highway LAN site. For reference, under the Corporate Average Fuel Economy standards, the state of California aims to achieve a fleet-wide average fuel economy of 54.5 miles per gallon by the year 2025 (US EPA, 2012), corresponding to a 35% decrease in emissions relative to the 35.5 miles per gallon economy of 2012–2016 model year vehicles. Assuming a steady decrease in emissions of 3.5% per year, one BEACO$_2$N site is therefore sufficiently sensitive to detect such a trend with 68% confidence in as little as 3 years. By leveraging observations from multiple independent sites, even greater confidence and/or shorter timescales could be achieved."

*P7 L17: The assumption that plumes can be detected within an urban area should be supported, e.g., by citations. At scales below 1 km² it seems that street canyon effects, building disturbances, etc. could play an important role and hinder the application of concepts such as "plumes," see, e.g., Lietzke and Vogt (2013; https://www.sciencedirect.com/science/article/pii/ S1352231013002069) that also investigated traffic emissions at street scale.*

Previous work using BEACO$_2$N measurements has provided preliminary evidence that plume-like events (if not "plumes" in the strictest sense) can be detected at this scale in urban areas (Kim et al., 2018). These plume-like events are not necessarily representative of a single vehicle's tailpipe, for example, but are nonetheless characterized by a sharp, distinct enhancement above background

concentrations and have been shown to be correlated with average emission factors expected for a given vehicle fleet. We have updated the text to include a reference to this important proof-of-concept study:

"Prior studies have demonstrated a methodology for detecting plume-like events in the $BEACO_2N$ $NO_x$ and CO observations (Kim et al., 2018), and the ratio of these species to $CO_2$ provides a unique signature for each different $CO_2$ source (e.g., Ban-Weiss et al., 2008; Harley et al., 2005; Lopez et al., 2013; Nathan et al., 2018; Turnbull et al., 2015), allowing subsets of the data record to be directly attributed to specific (e.g., mobile) source types and allowing the relationship between these specific activities and $CO_2$ mixing ratios to be derived more precisely."

*P8 L5: I would suggest reconsidering the wording here, especially as you refer to MRV earlier in the manuscript. This work strongly supports the conclusions of Turner et al. (2016), but it seems you have validated and not verified them.*

We appreciate the referee's attention to detail in this case and have updated the text accordingly:

*"This work thus provides an important data-based validation of the conclusions of Turner et al.'s theoretical analysis."*

*References:*

Kim, J., Shusterman, A. A., Lieschke, K. J., Newman, C., and Cohen, R. C.: The BErkeley Atmospheric $CO_2$ Observation Network: field calibration and evaluation of low-cost air quality sensors, Atmos. Meas. Tech., 11, 1937–1946, doi:10.5194/amt-11-1937-2018, 2018.

Shusterman, A. A., Teige, V. E., Turner, A. J., Newman, C., Kim, J., and Cohen, R. C.: The BErkeley Atmospheric $CO_2$ Observation Network: initial evaluation, Atmos. Chem. Phys., 16, 13449–13463, doi:10.5194/acp-16-13449-2016, 2016.

---

## Author Response (AR2)

We thank the co-editor for their ongoing work to provide comments leading to the improvement of our manuscript. Our responses to these comments are detailed below:

*Thank you very much for your revised manuscript.*

*I have a few additional requests/suggestions which I would like you to consider. All page and line numbers are with respect to the track changed manuscript in the Authors response file.*

*I also noted that fuel efficiency is given in miles per gallon. Please use SI units for this (e.g., 1/100 km or km liter$^{-1}$). Similar comment with respect to temperature (e.g., ppm $CO_2$/K not /$^o$C).*

We have updated the manuscript to include SI units exclusively.

*P3 L26: I'm not convinced that this will satisfy the reader. Of course, after long averaging, the error of the mean reduces. I would however be very careful to call this an uncertainty. After longer averaging, the instrumental uncertainty decreases but the influence of atmospheric variability increases. This should in some way be mentioned/discussed and also, I suggest to give a few examples of instrumental uncertainties after typical averaging times used in the manuscript (e.g., 5 min used in the correlation analysis). I also suggest to include a discussion of long-term drift as suggested by the reviewer. I find the answer in the reply to the reviewers rather vague. And it is one of those examples where other papers need to be read to understand the procedure.*

We have revised the text to address the co-editor's comments regarding the effect of averaging. The full long-term drift correction procedure is quite involved and well described in a prior publication. We have added our response to the referee to the main text of the manuscript:

"The processed 1-minute averages are assumed to have an instrumental uncertainty of less than ±4 ppm. The longer averaging timescales used hereafter reduce the error of the mean (e.g., ±1.8 ppm at 5-minute resolution, ±0.5 ppm at hourly resolution, ±0.06 ppm for a given hour of the day over an entire season, etc.), although the concomitant increase in the influence of atmospheric variability cannot be quantified. Any long-term drift in the sensors is accounted for via a combination of periodic (i.e., every 12–24 months) laboratory recalibration and a post hoc data treatment based on an independent reference site in the network domain. This procedure allows us to confidently compare measurements taken multiple years apart, thus enabling inter-annual changes in $CO_2$-related phenomena to be monitored. The exact details of the calibration and post hoc data treatment are provided in Shusterman et al. (2016)."

*Reviewer comment on important of other sources (and sinks): Please include a statement on this in the manuscript, reflecting your reply.*

We have updated the manuscript text in two places to reflect our response to the referee regarding (1) other anthropogenic sources and (2) the biosphere:

"By definition, we expect these local signals to represent a unique combination of emission sources and atmospheric dynamics specific to a given site. Here we endeavor to determine whether measurements of local $CO_2$ enhancements can be used to monitor a single urban emission source,

despite the complex landscape of $CO_2$ sources and sinks present within the study domain. We choose to focus on mobile $CO_2$ emissions as these are estimated to comprise approximately 40% of the San Francisco Bay Area's annual $CO_2$ emissions (Claire et al., 2015). This is the largest source sector in the $CO_2$ emission inventory and likely to represent an even larger fraction of emissions originating from within the urban core, where the next largest source sectors (industrial/commercial and electricity/co-generation) are less abundant."

"In addition to this first-order sensitivity to vehicle emissions at the near-roadway LAN site, we find that relatively subtle emission changes can also be detected using nodes stationed greater distances from the highway by controlling for the confounding impacts of dispersion and the biosphere. To do so, we decompose the $CO_2$ signals into terms that represent the influence of meteorology (which is correlated with both dispersion and biosphere activity) and emissions separately via a multiple linear regression approach analogous to that described by de Foy (2018)."

*Reviewer comment on the use of a single highway traffic count: Please include a statement on this in the manuscript, reflecting your reply.*

We have added the following text to the manuscript to reflect our reply to the referee:

"When we examine the relationship between these multiple linear regression coefficients and morning traffic flow as we did at LAN (Fig. 10), we again find positive correlations. This is an interesting result, given that the traffic flow measured on a single highway likely provides only a first order approximation of the total traffic emissions influencing a single $CO_2$ monitor, especially those situated at greater distances from said highway, which may be sensitive to additional highways, as well as local roads. Although the predominance of a single highway's emissions (or at least its correlation with those from other sources) is not a necessary condition of our MLR analysis, the strong positive correlations we observe suggest that this methodology may nonetheless be useful in monitoring emissions from individual highways such as these."

*P6 L24: Please be more precise: in local $CO_2$ mole fraction enhancements? Emissions?*

We have revised the text to clarify that this statement refers to an analysis of the $CO_2$ mole fraction enhancements:

*"An alternative analysis using traffic density–obtained by dividing the traffic flow by the average vehicle speed–yields almost identical results (Fig. S5), revealing a factor of 2 increase in local $CO_2$ mole fraction enhancements during congestion (high traffic flow/density) relative to free-flowing conditions (low traffic flow/density), similar to that observed by a previous on-road mobile monitoring study by Maness et al. (2015)."*

*Reviewer comment on intercept value of MLR: Please explain in more detail where the intercept is explained in the revised manuscript and what the physical meaning of this intercept is.*

After multiple comments expressing confusion about its derivation and significance, we have elected to remove the discussion of the MLR intercept value from the manuscript.

*Reviewer comment regarding P7 L4 (original manuscript): I'm not convinced that this answer satisfies the request of the reviewer to provide a robust explanation of how this value is derived. (I guess it is derived by dividing the 11% slope uncertainty by 3.5% year$^{-1}$ expected increase in fuel efficiency?)*

The co-editor has inferred the correct interpretation of our derivation; we have updated the manuscript text to hopefully clarify this point further:

"The 1σ uncertainty in the slopes (i.e., the 68% confidence interval, assuming a Gaussian error distribution) is thus found to be 11–30%, indicating that analysis of a single site could be used to detect as small as 11% changes in average emissions per vehicle, an improvement upon the 17% slope uncertainty calculated for the near-highway LAN site. For reference, under the Corporate Average Fuel Economy standards, the state of California aims to achieve a fleet-wide average fuel economy of 23.2 km per liter by the year 2025 (US EPA, 2012), corresponding to a 35% decrease in emissions relative to the 15.1 km per liter economy of 2012–2016 model year vehicles. Assuming a steady decrease in emissions of 3.5% per year, an 11% decrease would be achieved after approximately 3 years, showing that one BEACO$_2$N site is therefore sufficiently sensitive to detect such a trend with 68% confidence in as little as 3 years."

*P10 L5: Please specify what 1 ppm means. Over which time period?*

We have revised the text to clarify the conditions of the Turner et al. (2016) study:

[revised manuscript text omitted]
 Mondays (orange solid line), Fridays (blue dashed line), and Saturdays (black dotted line) between 15 February 2017 and 15 February 2018.**